# PowerFlow: Unlocking the Dual Nature of LLMs via Principled Distribution Matching

**Ruishuo Chen** [1]  **Yu Chen** [1]  **Zhuoran Li** [1]  **Longbo Huang** [1]

## Abstract

Unsupervised Reinforcement Learning from Internal Feedback (RLIF) has emerged as a promising paradigm for eliciting the latent capabilities of Large Language Models (LLMs) without external supervision. However, current methods rely on heuristic intrinsic rewards, which often lack a well-defined theoretical optimization target and are prone to degenerative biases. In this work, we introduce PowerFlow, a principled framework that reformulates unsupervised fine-tuning as a distribution matching problem. By casting GFlowNet as an amortized variational sampler for unnormalized densities, we propose a length-aware Trajectory-Balance objective that explicitly neutralizes the structural length biases inherent in autoregressive generation. By targeting $\alpha$-power distributions, PowerFlow enables the directional elicitation of the dual nature of LLMs: sharpening the distribution ($\alpha > 1$) to intensify logical reasoning, or flattening it ($\alpha < 1$) to unlock expressive creativity. Extensive experiments demonstrate that PowerFlow consistently outperforms existing RLIF methods, matching or even exceeding supervised GRPO. Furthermore, by mitigating over-sharpening in aligned models, our approach achieves simultaneous gains in diversity and quality, shifting the Pareto frontier in creative tasks.

## 1. Introduction

The ongoing debate regarding whether reinforcement learning (RL) can empower Large Language Models (LLMs) to transcend the capabilities inherent in their base models has sparked renewed interest in the latent potential embedded within pre-trained weights (Yue et al., 2025; Shao et al., 2025; Liu et al., 2025b). Consequently, significant research effort has been directed toward eliciting these capabilities without relying on labeled trajectories or external reward signals. In this landscape, Reinforcement Learning from Internal Feedback (RLIF) has emerged as a dominant paradigm. By employing intrinsic rewards derived from self-certainty (Zhao et al., 2025; Prabhudesai et al., 2025; Li et al., 2025) or ensemble-based consistency (Zuo et al., 2025; Zhang et al., 2025c), RLIF seeks to induce a process of self-evolution, guiding models toward more confident and consistent outputs to bolster reasoning performance.

Despite their empirical successes, existing RLIF methods predominantly rely on handcrafted reward designs that heuristically specify optimization directions. Lacking a principled theoretical objective, these approaches are often susceptible to the unintended biases inherent in their specific reward formulations. As a result, models frequently suffer from various pathological behaviors—particularly during over-optimization (Ghimire et al., 2026)—such as distorted response lengths (e.g., collapse (Shafayat et al., 2025) or explosion (Zhao et al., 2025)), overconfidence (Zhang et al., 2025d), and mode collapse (He et al., 2026).

Recent research attributes reasoning gains from RL post-training to distributional sharpening (Shao et al., 2025; Yue et al., 2025; Karan & Du, 2025), characterizing LLM self-improvement, including existing RLIF paradigms, as the implicit concentration of probability mass relative to the base distribution (Huang et al., 2025; He et al., 2026). Conversely, empirical evidence also indicates that over-sharpened distributions, often a byproduct of alignment, can stifle generative diversity and creative expression (Yang et al., 2025; West & Potts, 2025; Zhang et al., 2025b).

Inspired by these insights, we propose **PowerFlow**, a principled framework that reformulates unsupervised fine-tuning as a *distribution matching problem*. Moving beyond heuristic rewards, we target the $\alpha$-*power distribution* of the base model: $p_\alpha(y|q) \propto p_{\text{base}}(y|q)^\alpha$ (also known as the $\alpha$-order escort distribution (Beck & Schlögl, 1993)). This target is uniquely motivated by its ability to modulate entropy while strictly preserving the intrinsic structural features and relative mode rankings of the base distribution. We treat the power exponent $\alpha$ as a controllable knob to directionally

[1]Institute for Interdisciplinary Information Sciences, Tsinghua University, Beijing, China. Correspondence to: Longbo Huang <longbohuang@tsinghua.edu.cn>.

*Proceedings of the 43rd International Conference on Machine Learning*, Seoul, South Korea. PMLR 306, 2026. Copyright 2026 by the author(s).

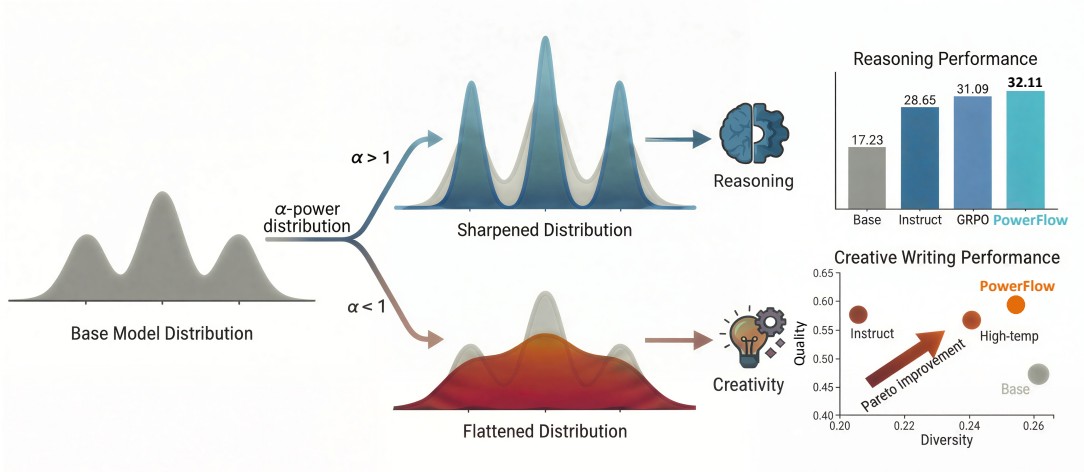

*Figure 1.* **Illustration of the PowerFlow framework for directional capability elicitation.** By matching the length-aware $\alpha$-power distribution, PowerFlow can either *sharpen* the distribution ($\alpha > 1$) to enhance logical reasoning or *flatten* it ($\alpha < 1$) to restore latent creativity. The right panels illustrate significant performance gains and a clear Pareto improvement over existing baselines.

elicit the dual nature of LLMs: sharpening the distribution ($\alpha > 1$) to concentrate mass on latent reasoning paths, or flattening it ($\alpha < 1$) to release the creative potential typically suppressed in aligned models. We provide a conceptual overview of the PowerFlow framework in Figure 1.

To operationalize this distribution matching, we build upon Generative Flow Network (GFlowNet) (Bengio et al., 2021; Malkin et al., 2022), a principled paradigm for learning stochastic policies that sample proportional to unnormalized densities. However, applying GFlowNets to the autoregressive structure of LLMs reveals a critical challenge: the exponential decay of trajectory probabilities. Without explicit correction, standard distribution matching objectives are often dominated by length-related variance rather than semantic density, leading to pathological length collapse during sharpening ($\alpha > 1$) or repetitive explosion during flattening ($\alpha < 1$) (Zhao et al., 2025; Shafayat et al., 2025).

To bridge this gap, we introduce a length-aware Trajectory-Balance (LA-TB) objective tailored for unsupervised LLM alignment. By reparameterizing the partition function into an amortized token-level energy term, we enable optimization on a length-normalized energy surface. This formulation effectively decouples the training gradient from response length and neutralizes the structural length bias inherent in autoregressive generation. As illustrated in Figure 3, this principled derivation allows PowerFlow to achieve stable optimization and monotonic performance gains that raw distribution matching objectives fail to maintain.

Extensive evaluations across various models and benchmarks reveal that PowerFlow ($\alpha > 1$) consistently outperforms current RLIF methods, matching or even surpassing the performance of supervised GRPO (Shao et al., 2024)

while preserving the diversity of reasoning paths. Furthermore, when applied to instruction-tuned models, PowerFlow ($\alpha < 1$) yields simultaneous diversity and quality gains in creative writing, successfully unlocking the latent creativity that is typically suppressed during the alignment process.

**Code Availability.** Code is available at https://github.com/Chenruishuo/PowerFlow.

Our contributions are summarized as follows:

- **PowerFlow Framework**: We propose a principled framework that reformulates unsupervised LLM fine-tuning as a distribution matching problem. By targeting the $\alpha$-power distribution, PowerFlow provides a unified theoretical foundation to directionally elicit the model's dual nature—enhancing reasoning or restoring creativity—via a single controllable parameter $\alpha$.

- **Length-Aware Objective**: We derive a length-aware Trajectory-Balance objective that neutralizes the exponential length bias inherent in autoregressive generation. This formulation enables stable, principled distribution alignment on a length-normalized energy surface, preventing the degenerative length collapse common in heuristic RLIF methods.

- **Dual-Nature Activation**: Extensive experiments demonstrate that PowerFlow ($\alpha > 1$) consistently outperforms existing RLIF methods and matches or exceeds supervised GRPO in reasoning accuracy. Conversely, PowerFlow ($\alpha < 1$) successfully restores the latent creativity of aligned models, achieving simultaneous gains in both output diversity and quality.

## 2. Related Works

RLIF (Zhao et al., 2025) elicits latent reasoning by replacing external reward models with intrinsic signals. Representative choices include self-certainty (Zhao et al., 2025), token entropy (Prabhudesai et al., 2025), generation probabilities (Li et al., 2025), semantic entropy (Zhang et al., 2025c), and majority voting (Zuo et al., 2025; Shafayat et al., 2025). Lacking a principled target, these handcrafted rewards exhibit recurring failures: overconfidence on instruct models (Zhang et al., 2025d), entropy-deflating shortcuts (Ghimire et al., 2026; He et al., 2026), majority-voting reward hacking (Shafayat et al., 2025), and probability-induced length collapse (Zhao et al., 2025; He et al., 2026).

A parallel line interprets RLVR gains (Shao et al., 2024; DeepSeek-AI et al., 2025) as "distribution sharpening" (Yue et al., 2025; Shao et al., 2025). Existing RLIF analyses in this lens (Huang et al., 2025; He et al., 2026) stay largely observational and overlook structural length bias. Karan & Du (2025) attain such sharpening via MCMC on the $\alpha$-power target, but at prohibitive inference cost. PowerFlow amortizes principled $\alpha$-power matching into training. Its length-aware Trajectory-Balance objective neutralizes structural length bias and elicits reasoning ($\alpha>1$) or creativity ($\alpha<1$) under single-pass decoding (extended in Appendix A).

## 3. PowerFlow: Unsupervised Fine-Tuning via Distribution Matching

### 3.1. The Distribution Matching Formulation

In the unsupervised fine-tuning of Large Language Models (LLMs), we operate in the absence of external reward signals or ground-truth trajectories. Our primary information source is the base model distribution $p_{\text{base}}(y|q) = \prod_{t=1}^{T} p_{\text{base}}(y_t|q, y_{<t})$, where $q$ is a given query and $y$ is the generated response. Consequently, any unsupervised training scheme can be conceptualized as a mechanism for redistributing the initial probability mass by leveraging information intrinsic to the base model itself.

In this work, we introduce **PowerFlow**, a framework that returns to the probabilistic essence of the problem by reformulating unsupervised fine-tuning as a *distribution matching problem*. Our goal is to minimize the divergence between the fine-tuned policy $\pi_\theta$ and a target distribution $p_{\text{target}}$. We primarily focus on the *$\alpha$-power distribution*, a self-derived and non-heuristic transformation of the base model:

$$p_\alpha(y|q) = \frac{p_{\text{base}}(y|q)^\alpha}{Z(q, \alpha)}, \tag{1}$$

where $Z(q, \alpha) = \sum_y p_{\text{base}}(y|q)^\alpha$ is the partition function. Unlike handcrafted rewards, the $\alpha$-power distribution, widely known as the escort distribution in statistical mechanics (Beck & Schlögl, 1993), provides a principled reshaping

of $p_{\text{base}}$ that modulates entropy while strictly preserving its relative probability rankings and mode structure via monotonicity. This ensures that the fine-tuning remains inherently grounded in the model's pre-trained knowledge, effectively circumventing the biased distributional drift often associated with heuristic rewards. Under the PowerFlow framework, we treat $\alpha$ as a controllable knob to directionally elicit the model's dual nature:

**Reasoning Elicitation ($\alpha > 1$):** Inspired by research linking reasoning gains to distribution sharpening (Yue et al., 2025; Karan & Du, 2025), we expect that matching $p_\alpha$ with $\alpha > 1$ will enhance reasoning accuracy. This objective is grounded in the principle of verification-generation asymmetry, supported by both computational complexity theory (Cook, 1971) and empirical studies (Weng et al., 2023; Yuan et al., 2024). This asymmetry, where recognizing a correct solution is easier than generating one, suggests that the model harbors substantial "hidden knowledge" (Hinton et al., 2015) that is not fully manifested during default generation. We hypothesize that a significant performance-capability gap exists due to the base model's relatively flat distribution. Sharpening thus acts as a mechanism to bridge this gap by concentrating probability mass on latent, high-quality reasoning paths, thereby increasing the efficiency of surfacing correct trajectories during standard decoding. Indeed, we theoretically demonstrate that the empirically effective majority-voting based RLIF can be formalized as an implicit mechanism for extreme distribution sharpening, effectively driving the policy towards the dominant mode (see Theorem F.1 in Appendix F).

**Creativity Release ($\alpha < 1$):** When $\alpha < 1$, the target distribution is flattened, redistributing probability mass toward the long-tail regions. While applying such flattening to a raw base model may lead to incoherent outputs due to its high intrinsic entropy, it serves a distinct purpose for aligned models. Recent studies observe that alignment procedures, such as RLHF (Ouyang et al., 2022), frequently induce excessive distribution sharpening, which suppresses the inherent creativity and diversity of the original base model (West & Potts, 2025; Yang et al., 2025; Lanchantin et al., 2025). Specifically, Zhang et al. (2025b) formalize the "typicality bias" inherent in reward models, asserting that aligned models implicitly sample from a $\alpha(> 1)$-power distribution of the reference model. This results in a pathological preference for high-probability, typical responses while discarding creative alternatives. We hypothesize that these creative capabilities remain latent within sub-peak probability regions. By matching a flattened distribution, PowerFlow facilitates the recovery of these suppressed expressive capabilities, effectively countering typicality bias to enable a more diverse and creative generation space.

Ultimately, PowerFlow transforms unsupervised fine-tuning

from a heuristic pursuit of rewards into a principled optimization task, where $\alpha$ serves as a precise control mechanism to navigate the model's latent capability space without the brittleness of manual reward specification. To this end, we first frame GFlowNet as an amortized solution to this distribution matching problem from the perspective of variational inference. We then derive a length-aware Trajectory-Balance objective designed to neutralize the structural length bias inherent in autoregressive generation, thereby ensuring stable and robust optimization. The complete algorithmic workflow is illustrated in Figure 2.

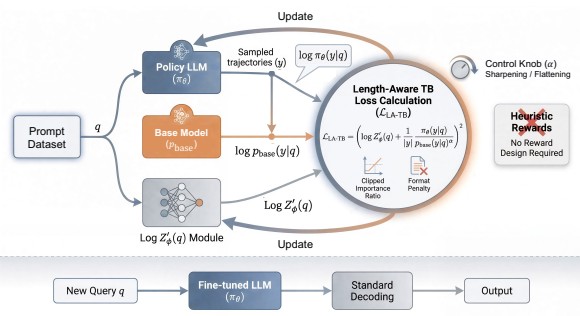

*Figure 2.* **The PowerFlow framework.** During training (top), the policy $\pi_\theta$ and $\log Z'_\phi$ module are optimized via the LA-TB objective to match the $\alpha$-power distribution of the base model while neutralizing length bias. The control knob $\alpha$ enables directional elicitation: sharpening ($\alpha > 1$) for reasoning or flattening ($\alpha < 1$) for creativity. The inference pipeline (bottom) remains standard.

## 3.2. GFlowNets as Amortized Samplers

A fundamental challenge in matching the target distribution $p_{\text{target}}(y|q)$ is that its partition function, $Z(q) = \sum_y \tilde{p}_{\text{target}}(y|q)$, is computationally intractable due to the combinatorial explosion of the response space $\mathcal{Y}$. From the perspective of Variational Inference (VI), this intractability can be bypassed by expressing the reverse KL divergence via a variational surrogate:

$$\mathbb{D}_{\text{KL}}\big(\pi_\theta\|p_{\text{target}}\big) = \mathbb{E}_{y\sim\pi_\theta}\left[\log\frac{\pi_\theta(y|q)}{\tilde{p}_{\text{target}}(y|q)}\right] + \log Z(q).$$
(2)

Since $\log Z(q)$ is constant with respect to the policy parameters $\theta$, minimizing the reverse KL divergence is equivalent to minimizing the expectation term, which serves as a variational upper bound.

Generative Flow Networks (GFlowNets) (Bengio et al., 2021) provide a principled framework for this variational setting by learning a policy that acts as an amortized sampler for unnormalized densities. Zimmermann et al. (2023) formally established that the *Trajectory Balance* (TB) objective (Malkin et al., 2022) acts as a specialized variational surrogate for minimizing Eq. (2). Intuitively, the forward policy $P_F$ is the actual generation policy from which one

samples at inference time to obtain trajectories from the target distribution, while the backward policy $P_B$ serves only as a training-time auxiliary that disambiguates parent states in the underlying DAG.

**Proposition 3.1** (Zimmermann et al. (2023)). *For a trajectory $\tau = (s_0, s_1, \ldots, s_T)$, define the Trajectory Balance objective as:*

$$\mathcal{L}_{TB}(\theta, \phi, \psi; \tau) = \left(\log\frac{Z_\phi \prod_{t=0}^{T-1} P_F(s_{t+1}|s_t; \theta)}{\tilde{p}_{target}(s_T) \prod_{t=1}^{T} P_B(s_{t-1}|s_t; \psi)}\right)^2.$$
(3)

*If the learned partition function $Z_\phi$ is optimized to its equilibrium, the expected gradient of $\mathcal{L}_{TB}$ with respect to the forward policy parameters $\theta$ satisfies:*

$$\mathbb{E}_{\tau\sim P_F}\left[\nabla_\theta\mathcal{L}_{TB}(\theta, \phi, \psi; \tau)\right] = 2\nabla_\theta\mathbb{D}_{KL}\big(P_F(\tau; \theta)\|p_{target}(\tau)\big).$$
(4)

In the context of LLMs, the autoregressive generation process naturally forms a tree-structured Directed Acyclic Graph (DAG), where each state $s_t$ corresponds to a prefix $y_{<t}$. Since each state in a tree has a unique parent, the backward policy simplifies to $P_B(y_{<t}|y_{\leq t}, q) = 1$ for all valid transitions. Identifying the forward policy $P_F$ as our model $\pi_\theta$, the TB loss for a given query $q$ and response $y$ reduces to a form tailored for autoregressive models:

$$\mathcal{L}_{\text{TB}}(\theta, \phi; q, y) =$$
$$\left(\log Z_\phi(q) + \sum_{t=1}^{T}\log\pi_\theta(y_t|y_{<t}, q) - \log\tilde{p}_{\text{target}}(y|q)\right)^2.$$
(5)

This formulation transforms the distribution matching problem into an RL-style on-policy optimization task:

$$\min_{\theta, \phi}\mathcal{J}(\theta, \phi) = \mathbb{E}_{q\sim\mathcal{D}}\left[\mathbb{E}_{y\sim\pi_\theta(\cdot|q)}\left[\mathcal{L}_{\text{TB}}(\theta, \phi; q, y)\right]\right].$$
(6)

By optimizing Eq. (6), $\pi_\theta$ learns to align its sequence-level probabilities with $\tilde{p}_{\text{target}}$, while $Z_\phi(q)$ amortizes the estimation of the normalization constant to reduce gradient variance. Further implementation and training details are provided in Appendix B.1.

## 3.3. Length-Aware PowerFlow Objective

Autoregressive generation in LLMs is inherently plagued by structural length bias. Specifically, the log-probability of a trajectory, $\log p(y|q) = \sum_{t=1}^{|y|}\log p(y_t|y_{<t}, q)$, is approximately negatively linear with respect to the sequence length $|y|$. Consequently, a naive distribution matching objective is often dominated by sequence length rather than semantic density. For instance, when targeting an $\alpha$-power distribution with $\alpha > 1$ (sharpening), the model tends to exploit the path probability by producing excessively short,

trivial sequences. Conversely, when $\alpha < 1$ (flattening), the model is prone to generating repetitive, deterministic long sequences to accumulate probability mass. Furthermore, the extreme sensitivity of path probabilities to $|y|$ causes the gradient of the partition function $Z_\phi$ to exhibit massive variance, severely destabilizing the optimization process.

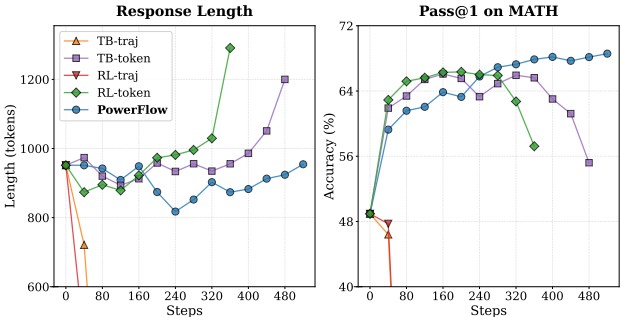

*Figure 3.* **Stability analysis of distribution matching strategies.** Matching the trajectory-level $\alpha$-power distribution via standard TB or RL objectives (*-traj*) leads to rapid length collapse. Token-level normalization (*-token*) initially improves performance but eventually decays due to the exploitation of repetitive tokens. PowerFlow maintains both stable response length and superior reasoning accuracy (pass@1 on MATH) throughout training.

As illustrated in Figure 3, directly matching the $\alpha(> 1)$-power distribution using either the Trajectory Balance (TB-traj) or RL-based KL-regularized objectives (RL-traj) leads to an immediate and pathological collapse of response length. We include the RL-based formulation as a baseline because the standard RL objective with KL regularization, $\max_\pi \mathbb{E}_{y \sim \pi}[r(y)] - \beta \mathbb{D}_{\mathrm{KL}}(\pi \| \pi_{\mathrm{base}})$, theoretically yields an optimal policy $\pi^*(y|q) \propto \pi_{\mathrm{base}}(y|q) \exp(r(y)/\beta)$. By setting the intrinsic reward to the base model's log-probability, $r(y) = \log p_{\mathrm{base}}(y|q)$, the target becomes an $\alpha$-power distribution where $\alpha = 1 + 1/\beta$.

To counteract length bias, a common heuristic is to optimize the average token log-probability, $\frac{1}{|y|} \log p_{\mathrm{base}}$. While these token-level variants, TB-token and RL-token, exhibit initial performance gains, Figure 3 reveals a subsequent decay. This failure stems from a fundamental distortion of the target distribution's structural integrity. By optimizing for average token-level confidence, these methods reshape the energy surface such that local probability mass is decoupled from global semantic coherence. Consequently, the model exploits repetitive and meaningless tokens to artificially lower the average energy, effectively eroding the learned semantic structure to inflate likelihood metrics through repetitive generation. These observations underscore that both naive RL and standard GFlowNet objectives are profoundly susceptible to reward and structural biases, necessitating a more principled approach to distribution alignment.

To bridge the gap between principled distribution matching and the non-stationary length distributions of LLMs, we

introduce a structural reparameterization of the Trajectory-Balance objective. Standard GFlowNets (Malkin et al., 2022) typically treat the partition function $Z_\phi(q)$ as a prompt-dependent scalar, a formulation that is ill-conditioned for autoregressive sequences where probabilities decay exponentially with length. We instead reformulate the normalization constant as a *length-aware energy term*, $Z_\phi(q, y) = (Z'_\phi(q))^{|y|}$, which effectively projects the optimization onto a *length-normalized energy surface*. By further normalizing the log-trajectory mismatch by $|y|$, our LA-TB objective ensures that the optimization gradient remains scale-invariant across varying sequence lengths:

$$\mathcal{L}_{\mathrm{LA\text{-}TB}}(\theta, \phi; q, y) = \left( \log Z'_\phi(q) + \frac{1}{|y|} \log \frac{\pi_\theta(y|q)}{\tilde{p}_{\mathrm{target}}(y|q)} \right)^2. \tag{7}$$

This objective converges to a length-normalized target distribution:

$$\pi^*(y|q) \propto \frac{\tilde{p}_{\mathrm{target}}(y|q)}{Z'_\phi(q)^{|y|}}. \tag{8}$$

While Eq. (8) does not preserve the full global ranking of $\tilde{p}_{\mathrm{target}}$ across arbitrary lengths, the resulting deformation is highly structured rather than arbitrary. Writing $\lambda_q := \log Z'_\phi(q)$, the length-aware distribution takes the form $\pi^*(y|q) \propto \tilde{p}_{\mathrm{target}}(y|q) e^{-\lambda_q|y|}$, which is a one-dimensional exponential tilt by the sufficient statistic $|y|$. This structure admits two clean guarantees.

**Proposition 3.2** (LA-TB is the I-projection / minimum-distortion length correction)**.** *Fix a prompt $q$ and an expected length budget $\ell_q$. Among all distributions $\pi(\cdot|q)$ with $\mathbb{E}_\pi[|y|] = \ell_q$, the unique minimizer of $\mathbb{D}_{\mathrm{KL}}(\pi(\cdot|q) \| \tilde{p}_{target}(\cdot|q))$ has the form $\pi^\star(y|q) \propto \tilde{p}_{target}(y|q) e^{-\lambda_q|y|}$. Hence Eq. (8) is exactly the I-projection of $\tilde{p}_{target}$ onto the family of length-calibrated distributions: it is the least-distorting modification of $\tilde{p}_{target}$ that aligns its expected response length with the model.*

**Proposition 3.3** (Global KL distortion is second-order in $\lambda_q$)**.** *Assume the moment generating function $M_q(t) := \mathbb{E}_{y \sim \tilde{p}_{target}(\cdot|q)}\left[ e^{t|y|} \right]$ is finite on $|t| < \delta$ for some $\delta > 0$, and let $A_q(\lambda) := \log M_q(-\lambda)$ denote the cumulant generating function of $|y|$ under $\tilde{p}_{target}(\cdot|q)$, so that $A_q$ is real-analytic on $(-\delta, \delta)$ with $A''_q(0) = \mathrm{Var}_{\tilde{p}_{target}}(|y|)$. The global LA-TB distortion admits the exact identity $\mathbb{D}_{\mathrm{KL}}(\pi^* \| \tilde{p}_{target}) = \lambda_q A'_q(\lambda_q) - A_q(\lambda_q)$, and the Taylor expansion of $A_q$ at 0 yields*

$$\mathbb{D}_{\mathrm{KL}}\left( \pi^*(\cdot|q) \, \| \, \tilde{p}_{target}(\cdot|q) \right) = \frac{\lambda_q^2}{2} \mathrm{Var}_{\tilde{p}_{target}}(|y|) + O(|\lambda_q|^3). \tag{9}$$

*The global distortion induced by LA-TB is therefore controlled to second order by the target's response-length variance under the learned multiplier $\lambda_q$, and vanishes in the small-$\lambda_q$ regime where the base length statistics are already close to the calibrated budget.*

Together, Propositions 3.2 and 3.3 establish LA-TB *not* as a heuristic reward shaping, but as the minimum-distortion length correction whose global divergence from the ideal target is quantitatively controlled at second order in $\lambda_q$, with $\ell_q := \mathbb{E}_{\pi^*}[|y|]$ taken to be the induced equilibrium length (see Appendix E for the precise identification). An exact within-shell preservation result (the mode ranking of $\tilde{p}_{\text{target}}$ inside every fixed-length shell is preserved exactly), the exact pairwise log-margin formula, a top-$K$ preservation corollary, and all proofs are deferred to Appendix E. Empirically, on a trained Qwen2.5-Math-1.5B we measure the pairwise inversion rate of LA-TB relative to the ideal $\alpha$-power target and find IR $\approx 0.09$, i.e., roughly $91\%$ of $\alpha$-power rankings are preserved in practice, in line with the structural guarantees above. By operating within the space of amortized geometric mean probabilities, PowerFlow prioritizes semantic quality over sequence brevity or redundancy, effectively shifting the optimization focus toward the model's true latent capability space.

Finally, we instantiate the target as the $\alpha$-power distribution, $p_{\text{base}}(y|q)^{\alpha}$. To ensure instruction-following integrity and logical structure, we incorporate a format penalty $\psi(y)$. Specifically, $\psi(y)$ is set to a negative constant (e.g., $-0.5$) if the output fails to match a predefined pattern (e.g., the absence of \boxed{}), and $0$ otherwise. This yields the final PowerFlow objective:

$$\mathcal{L}_{\text{PowerFlow}} = w \cdot \left( \log Z'_\phi(q) + \frac{1}{|y|} \log \pi_\theta(y|q) \right. $$
$$\left. - \alpha \left[ \frac{1}{|y|} \log p_{\text{base}}(y|q) + \psi(y) \right] \right)^2 \tag{10}$$

where $w$ is the importance sampling ratio defined as:

$$w = \text{clip} \left( \frac{\pi_\theta(y|q)}{\pi_{\text{old}}(y|q)}, 1 - \epsilon, 1 + \epsilon \right)^{\text{detach}}. \tag{11}$$

The inclusion of $w$ ensures compatibility with off-policy fine-tuning, where trajectories are sampled from a behavior policy $\pi_{\text{old}}$. Following Zhu et al. (2025), we apply clipping to maintain training stability and prevent gradient collapse during iterative optimization. As evidenced by the robust training dynamics in Figure 3, the PowerFlow objective effectively circumvents these structural distortions, achieving sustained length stability and monotonic performance gains by preserving the principled $\alpha$-power density on a length-normalized surface.

# 4. Experiments

In this section, we evaluate the effectiveness of PowerFlow across two primary domains: complex logical reasoning and diverse creative writing. Following the experimental setup detailed in Section 4.1, we present our findings in two parts. First, Section 4.2 demonstrates that distribution sharpening ($\alpha > 1$) effectively intensifies reasoning performance across various model variants. Subsequently, Section 4.3 reveals that distribution flattening ($\alpha < 1$) restores the generative diversity typically suppressed in aligned models while simultaneously improving output quality. Together, these experiments illustrate that principled distribution matching serves as a robust mechanism for the directional elicitation of latent LLM capabilities without external supervision.

## 4.1. Experimental Setup

**Data and Training Configuration.** For reasoning tasks, we follow standard practices in the community (Hugging Face, 2025; Zhang et al., 2025a) by utilizing questions from the NuminaMath-CoT dataset (LI et al., 2024) for unsupervised training. Specifically, we employ a subset of 18,000 queries filtered by Zhang et al. (2025c) to exclude instances with excessive response length or potential answer leakage. Each query is appended with a prompt instructing the model to "think step by step" and provide the final answer within a \boxed{} environment. For creative writing tasks covering poem continuation, story generation, and joke writing (Lu et al., 2025), we select a training set of 300 prompts, drawn from the 500-prompt collection curated by Zhang et al. (2025b) and sourced from PoemHunter.com, BookMIA (Shi et al., 2024), and Reddit r/DadJokes (Reddit, 2023). All inputs are formatted using the models' official chat templates; detailed prompts and hyperparameters are provided in Appendix C and Appendix B.2, respectively.

**Models and Baselines.** To evaluate the generalizability of PowerFlow, we conduct experiments across several representative model families and scales, including the 1.5B, 3B, and 7B variants of the Qwen2.5 series, the domain-specific Qwen2.5-Math series, and the Llama-3.2 series.

For reasoning benchmarks, we compare PowerFlow against a comprehensive suite of baselines: (i) *Base*: The original un-finetuned model. (ii) *Low-temp*: The *Base* model using a reduced inference temperature ($T' = T/\alpha$) following Karan & Du (2025). (iii) *Instruct*: The official instruction-tuned version of the respective base model. (iv) *Format-only*: A variant of our framework using $\alpha = 1$, isolating the performance gains solely from improved answer extractability via the format penalty. (v) RLIF Methods: State-of-the-art unsupervised methods including *Intuitor* (self-certainty rewards) (Zhao et al., 2025), *EMPO* (semantic entropy rewards) (Zhang et al., 2025c), and *TTRL* (majority-voting rewards) (Zuo et al., 2025). (vi) *PowerSampling*: An inference-time baseline that samples from the $\alpha$-power distribution using Markov Chain Monte Carlo (MCMC) methods (Karan & Du, 2025). (vii) *One-shot EM*: A method that directly minimizes token-level entropy via unsupervised

optimization (Gao et al., 2025). (viii) *GRPO*: Group Relative Policy Optimization (Shao et al., 2024) trained on the NuminaMath-CoT dataset with external verifiable rewards, representing a supervised counterpart. For GRPO, we follow the DAPO (Yu et al., 2026) training paradigm and verl's (Sheng et al., 2025) recipe, using an answer-verification reward that checks whether the final answer inside \boxed{} matches the ground truth, together with asymmetric Clip-Higher to encourage exploration (which we retain when training PowerFlow as well). For reproducibility, we utilize the official open-source checkpoints for RLIF baselines, given the substantial computational overhead of full-scale fine-tuning. We discuss this choice in Appendix G. Our training setup mirrors the EMPO recipe established in Zhang et al. (2025c), which ensures a fair comparison against the released EMPO checkpoints.

For creative writing tasks, we compare PowerFlow ($\alpha = 0.5$) with: (i) *Instruct*: The default instruct-tuned model. (ii) *Base*: The original un-finetuned (pre-trained) model. (iii) *High-temp*: The *Instruct* model with an increased sampling temperature. (iv) *VS-Standard*: The "Verbalized Sampling" method (Zhang et al., 2025b), which improves diversity by adjusting prompts to elicit latent expressive variety.

**Evaluation.** For reasoning tasks, we report the mean accuracy across 16 independent samples (avg@16) per problem, with sampling temperature and top-$p$ fixed at 1.0. Our evaluation spans various mathematical reasoning benchmarks, including MATH500 (Hendrycks et al., 2021), OlympiadBench (He et al., 2024), AIME24 (LI et al., 2024), AIME25 (Zhang & Math-AI, 2025), and AMC23 (LI et al., 2024). We also include GPQA (diamond) (Rein et al., 2024) to assess natural scientific reasoning. Following Zeng et al. (2025), we extract answers within \boxed{} tags and employ a unified script to determine mathematical equivalence.

To assess creative writing, we utilize the remaining 200 prompts. Per prompt, we sample 30 independent responses at a temperature of 0.8 (excluding the *High-temp* baseline at 1.0) to ensure a robust estimation of semantic diversity and quality. To maintain consistency across phases, both training and evaluation employ official chat templates with concise, generation-restricting instructions. These constraints are designed to minimize conversational overhead, preventing distortion of probability calculations and ensuring that the computed densities strictly reflect the creative generation.

We quantify performance using three metrics: (i) *Semantic Diversity*: Following established practice (Shaib et al., 2024; Cann et al., 2023; Lu et al., 2025), defined as $1-\bar{s}$, where $\bar{s}$ is the mean pairwise cosine similarity of response embeddings from bge-small-en-v1.5 (Xiao et al., 2023). (ii) *Lexical Redundancy*: Measured by ROUGE-L scores among outputs following Shaib et al. (2024), where lower scores indicate higher variety. (iii) *Quality*: Evaluated via LLM-as-a-judge using Qwen3-plus, scoring responses based on rubrics from Creative Writing v3 (Paech, 2025) and Humor-Bench (Narad et al., 2025) (see Appendix C.2 for details).

## 4.2. Eliciting Reasoning via Distribution Sharpening

This section evaluates the efficacy of PowerFlow in eliciting latent reasoning capabilities through distribution sharpening ($\alpha > 1$). Based on preliminary tuning on Qwen2.5-Math-1.5B (detailed in Appendix D.1), we adopt a default sharpening parameter of $\alpha = 4$ for base models, aligning with the optimal configurations identified by Karan & Du (2025). Notably, for instruction-tuned models, we find that a lower value of $\alpha = 2$ is more effective; this is because these models have already undergone distribution sharpening during the alignment process, requiring less intensification to reach peak performance. To prioritize assessing our framework's generalizability across diverse architectures and scales, we adopt these values as robust defaults rather than pursuing exhaustive per-model optimization. While individual models possess unique intrinsic entropy profiles that may benefit from specialized $\alpha$-schedules, we leave the development of automated adjustment mechanisms for future work.

### 4.2.1. MAIN PERFORMANCE RESULTS

Table 1 summarizes the performance of PowerFlow across various model series and reasoning benchmarks. The results indicate that PowerFlow yields significant performance improvements that far exceed the gains achievable through simple temperature scaling (Low-temp), standard instruction tuning (Instruct) or targeted format regularization (Format-only). By performing principled distribution sharpening, PowerFlow effectively surfaces the latent reasoning capabilities inherent in the base models, consistently surpassing existing RLIF methods across all tested scales.

Notably, our unsupervised approach outperforms the supervised GRPO on three configurations—Qwen2.5-1.5B ($19.85_{\pm 0.28}$ vs. $18.13_{\pm 0.29}$), Qwen2.5-Math-1.5B ($34.30_{\pm 0.30}$ vs. $32.75_{\pm 0.31}$), and Llama-3.2-3B-Instruct ($22.88_{\pm 0.33}$ vs. $22.33_{\pm 0.33}$)—while achieving comparable results on Qwen2.5-Math-7B ($42.17_{\pm 0.27}$ vs. $42.38_{\pm 0.38}$). Subscripts denote one standard error of the mean (SEM) over the Average column, estimated by bootstrap over the 16 samples per problem; all improvements we claim over GRPO exceed $1\sigma$. These findings suggest that sharpening the distribution while preserving the structural integrity of the initial model provides a competitive elicitation mechanism that can rival, or even surpass, traditional reward-driven RL methods without the need for external labels or verifiers. We further analyze the underlying elicitation mechanism and provide the comparative Pass@$n$ performance in Appendix D.2.

*Table 1.* Model Performance Comparison (avg@16) across Benchmarks. Within each model block, horizontal lines separate unsupervised methods (top: *PowerFlow* and RLIF baselines) from reference models and supervised counterparts (bottom: *Instruct* and *GRPO*).

| | MATH500 | Olympiad | AIME24 | AIME25 | AMC23 | GPQA | **Average** |
|---|---|---|---|---|---|---|---|
| **Qwen2.5-1.5B** | | | | | | | |
| Base | 6.20 | 1.90 | 0.00 | 0.00 | 1.40 | 25.80 | 5.88 |
| Low-temp | 18.60 | 4.90 | 0.80 | 0.40 | 7.50 | 26.60 | 9.80 |
| Intuitor | 47.40 | 15.30 | 1.50 | 0.80 | 22.30 | 26.40 | 18.95 |
| **PowerFlow (Ours)** | 49.30 | 16.00 | 0.80 | 1.50 | 23.80 | 27.70 | **19.85** |
| Instruct | 34.90 | 10.00 | 0.80 | 0.00 | 13.60 | 28.00 | 14.55 |
| GRPO | 45.40 | 14.10 | 1.00 | 0.40 | 21.90 | 26.00 | 18.13 |
| **Qwen2.5-Math-1.5B** | | | | | | | |
| Base | 43.30 | 20.90 | 4.60 | 1.90 | 28.40 | 26.10 | 20.87 |
| Low-temp | 62.90 | 30.10 | 9.40 | 4.00 | 49.50 | 28.70 | 30.77 |
| Format-only | 65.70 | 30.10 | 5.60 | 5.00 | 47.0 | 26.10 | 29.92 |
| EMPO | 69.90 | 32.20 | 12.30 | 4.60 | 46.20 | 29.50 | 32.45 |
| **PowerFlow (Ours)** | 70.90 | 32.50 | 10.80 | 10.00 | 53.30 | 28.30 | **34.30** |
| Instruct | 71.50 | 33.50 | 10.20 | 6.00 | 49.40 | 26.40 | 32.83 |
| GRPO | 71.40 | 34.00 | 8.10 | 6.70 | 49.50 | 26.80 | 32.75 |
| **Llama-3.2-3B-Instruct** | | | | | | | |
| Base | 40.10 | 10.30 | 4.00 | 0.00 | 18.80 | 29.50 | 17.12 |
| Low-temp | 50.00 | 17.50 | 9.60 | 0.60 | 24.80 | 28.80 | 21.88 |
| Intuitor | 50.40 | 16.60 | 9.20 | 0.20 | 27.30 | 30.50 | 22.37 |
| **PowerFlow (Ours)** | 50.60 | 16.60 | 10.70 | 0.40 | 28.80 | 30.20 | **22.88** |
| GRPO | 50.10 | 17.20 | 11.20 | 0.00 | 25.00 | 30.50 | 22.33 |
| **Qwen2.5-Math-7B** | | | | | | | |
| Base | 46.70 | 22.30 | 12.30 | 4.20 | 34.50 | 29.70 | 24.95 |
| Low-temp | 69.00 | 34.00 | 23.10 | 7.70 | 47.70 | 34.10 | 35.93 |
| PowerSampling | 72.20 | 39.50 | 23.30 | 10.00 | 57.50 | 36.30 | 39.80 |
| TTRL | 80.40 | 39.60 | 21.70 | 11.90 | 58.80 | 34.70 | 41.18 |
| One-shot EM | 61.40 | 29.80 | 18.10 | 6.20 | 48.90 | 32.80 | 32.87 |
| EMPO | 79.30 | 41.70 | 15.80 | 12.30 | 60.20 | 36.00 | 40.88 |
| **PowerFlow (Ours)** | 78.10 | 40.10 | 20.00 | 14.40 | 63.40 | 37.00 | **42.17** |
| Instruct | 74.50 | 31.20 | 12.50 | 12.30 | 65.90 | 35.00 | 38.57 |
| GRPO | 78.40 | 42.50 | 22.70 | 12.90 | 63.40 | 34.40 | **42.38** |
| Qwen2.5-32B-Instruct | 79.70 | 43.10 | 13.10 | 9.40 | 60.30 | 44.10 | 41.62 |

### 4.2.2. PRESERVATION OF SOLUTION DIVERSITY

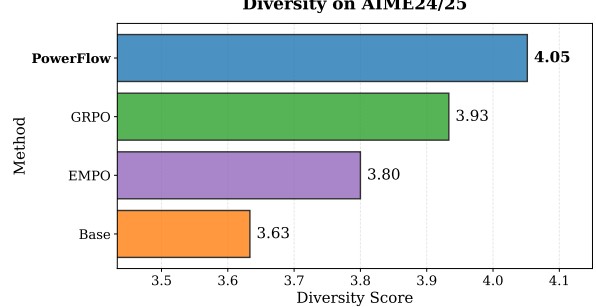

*Figure 4.* Comparison of solution diversity scores on AIME24/25. PowerFlow maintains superior strategy variety.

Recent studies highlight the potential for models to converge toward less diverse output patterns during RLIF (Zhang et al., 2025d; Ghimire et al., 2026; He et al., 2026). To investigate if PowerFlow inherits this vulnerability, we assess the diversity of reasoning paths on the AIME24 and AIME25 benchmarks. Using DeepSeek-V3.2 (Liu et al., 2025a) as an LLM-as-a-judge, we evaluate distinct solution strategies across 16 independent samples per problem, following the rubric in Appendix C.1.

As shown in Figure 4, PowerFlow achieves the highest diversity score (4.05), notably outperforming both EMPO (3.80) and supervised GRPO (3.93). This preservation of variety stems from the mathematical nature of the $\alpha$-power distribution matching objective. Unlike traditional RL objectives that may prioritize the exploitation of a single high-reward trajectory, the $\alpha$-power transformation rescales the entire density surface while strictly maintaining the relative rankings and multi-modal structure of the base model. Consequently, PowerFlow intensifies the probability of correct

reasoning paths across the entire latent landscape, allowing the model to elicit a broad spectrum of viable strategies rather than collapsing into a monolithic output pattern.

### 4.3. Restoring Creativity via Distribution Flattening

We further investigate the effectiveness of distribution flattening ($\alpha = 0.5$) in restoring the creative diversity of aligned models, which often suffer from mode collapse during instruction tuning. Figure 5 illustrates the averaged performance across four model families, mapping the Pareto frontier between generation quality and semantic diversity. Additional results regarding lexical redundancy and per-task breakdowns are provided in Appendix D.4 and Table 3.

As shown in Figure 5, the Instruct models (squares) maintain high quality but exhibit severely restricted diversity. This aligns with prior observations of "typicality bias," where alignment suppresses high-quality but atypical semantic paths in favor of predictable responses. Conversely, while *Base* models (circles) possess high latent creativity, their excessive entropy and poor instruction-following frequently lead to incoherent outputs, preventing the realization of their full generative potential. Other baselines fail to bridge this gap effectively: increasing the sampling temperature (High-temp, diamonds) improves diversity only at the expense of quality, while VS-Standard (triangles) degrades quality on models at the 7B scale or smaller due to its reliance on advanced instruction-following capabilities.

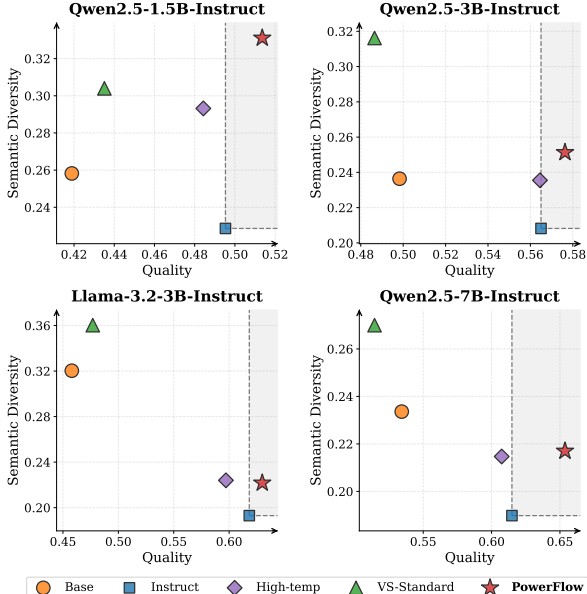

*Figure 5.* Quality vs. Semantic Diversity on creative writing tasks. The shaded region indicates the area of Pareto improvement relative to the *Instruct* baseline. PowerFlow (stars) consistently shifts the Pareto frontier outward across all model scales.

In contrast, PowerFlow (stars) amortizes the flattening of the energy surface to facilitate the exploration of high-quality, unconventional linguistic paths frequently suppressed by aligned distributions. By achieving superior semantic diversity and reduced lexical redundancy while simultaneously surpassing original *Instruct* models in quality, PowerFlow effects a Pareto-dominant shift. Notably, it stands as the only evaluated method capable of enhancing generation quality, demonstrating a successful preservation of robust instruction-following while releasing latent expressive potential. The resulting "best-of-both-worlds" synergy ensures the model remains highly controllable and stylistically sophisticated, yet retains the structural variety and creative entropy inherent in the pre-trained distribution. Such outcomes confirm that distribution flattening serves as a principled and effective mechanism for revitalizing the creative potential of aligned large language models.

## 5. Conclusion and Discussion

In this work, we introduced PowerFlow, a principled framework for unsupervised capability elicitation in LLMs. By framing fine-tuning as a distribution-matching problem, we demonstrated that the latent capabilities of LLMs can be directionally activated without external labels or verifiable rewards. Our results demonstrate that distribution sharpening ($\alpha > 1$) significantly enhances logical reasoning, rivaling or even surpassing supervised methods such as GRPO, while distribution flattening ($\alpha < 1$) restores the creative diversity frequently suppressed during standard instruction tuning. The success of PowerFlow is fundamentally rooted in its length-aware Trajectory Balance objective, which neutralizes the inherent length bias of autoregressive generation and ensures stable, non-degenerative optimization.

Beyond empirical gains, PowerFlow provides a robust diagnostic and optimization framework for investigating the distributional geometry of LLMs. Our findings suggest that pre-trained models possess remarkably sophisticated structural integrity, where RL-based fine-tuning often focuses more on optimizing distribution shapes than introducing novel knowledge. By shifting from heuristic reward engineering toward explicit distribution matching, we offer a transparent methodology generalizable beyond $\alpha$-power transformations to diverse distribution families. This paves the way for a unified unsupervised alignment paradigm where task-specific elicitation occurs by matching models to optimized target geometries. Future research on automating the discovery of these shapes and schedules could lead to a more efficient, theoretically grounded approach to developing versatile, specialized, and creative AI agents.

## Acknowledgements

This work was supported by the National Natural Science Foundation of China Grant 52494974.

## Impact Statement

This work presents a principled framework for the unsupervised elicitation of latent capabilities within Large Language Models (LLMs). By enabling the directional activation of reasoning and creativity through principled distribution matching, our framework mitigates the reliance on cost-intensive human-labeled datasets and external verifiable rewards, thereby democratizing the development of specialized and high-performing AI agents.

However, the capacity to manipulate a model's latent distributional modes warrants careful ethical consideration.. Distribution flattening, while intended to restore generative diversity, may inadvertently resurface or amplify harmful content previously suppressed through safety alignment protocols like RLHF. Conversely, aggressive sharpening could prioritize biased or suboptimal reasoning paths if the induced target distribution grants undue prominence to such modes. We therefore emphasize that the deployment of this framework should be fortified by robust safety guardrails and rigorous red-teaming to ensure that elicited capabilities remain socially beneficial and ethically grounded.

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

# A. Extended Related Works

**Reinforcement Learning from Internal Feedback (RLIF).** RLIF was pioneered by (Zhao et al., 2025) to facilitate unsupervised elicitation of reasoning capabilities by substituting external rewards—typically derived from pre-trained reward models or verifiers—with intrinsic feedback signals. This paradigm has catalyzed extensive research into diverse intrinsic reward mechanisms, including self-certainty (Zhao et al., 2025), token-level entropy (Prabhudesai et al., 2025), generation probabilities (Li et al., 2025), semantic entropy (Zhang et al., 2025c), and majority voting (Zuo et al., 2025; Shafayat et al., 2025).

Despite their empirical successes, most existing RLIF methods rely on handcrafted rewards that heuristically guide optimization without a principled target distribution. This lack of theoretical grounding leaves them vulnerable to the inherent biases of reward design, particularly in the later stages of policy optimization. For instance, entropy-based RLIF has been shown to induce overconfidence and performance degradation as training progresses, while exhibiting minimal impact on low-entropy instruct-tuned models (Zhang et al., 2025d). Recent work further reveals that such internal-confidence-based methods may trigger the generation of inappropriate, repetitive, and predictable patterns as a shortcut to artificially reduce entropy (Ghimire et al., 2026; He et al., 2026). Similarly, majority-voting rewards are susceptible to reward hacking, where models generate high-entropy, stochastic chains-of-thought to arrive at consistent but irrelevant answers (Shafayat et al., 2025). Furthermore, probability-based rewards are prone to length collapse as models exploit the higher joint probabilities of shorter sequences (Zhao et al., 2025; He et al., 2026). In contrast, PowerFlow eliminates heuristic reward biases by matching the principled $\alpha$-power distribution, while its length-aware objective prevents degenerative length collapse.

**The Distribution Sharpening Mechanism.** Following the remarkable success of Reinforcement Learning from Verifiable Rewards (RLVR) in enhancing LLM reasoning (Shao et al., 2024; DeepSeek-AI et al., 2025), significant effort has been devoted to understanding its underlying mechanisms. Critical insights from prior work (Yue et al., 2025; Shao et al., 2025) suggest that current RLVR training may not necessarily elicit entirely new reasoning patterns beyond the base model's intrinsic capacity. Instead, it operates through "distribution sharpening," a process of amplifying specific internalized reasoning paths to improve sample efficiency at low sampling rates (e.g., pass@1). While recent theoretical works have begun to analyze RLIF methods through this lens (Huang et al., 2025; He et al., 2026), they remain largely observational, lacking efficient practical sharpening methods and overlooking the structural length bias inherent in $\alpha$-power distributions. Notably, Karan & Du (2025) demonstrated that MCMC sampling from the sharpened $\alpha$-power distribution yields outstanding performance; however, its prohibitive inference cost and complex sampling logic render it impractical for general deployment. In contrast, PowerFlow amortizes the computational cost of distribution matching into the training phase, eliciting either logical reasoning via sharpening or expressive creativity via flattening under standard decoding at inference time.

# B. Implementation Details

In this section, we elaborate on the implementation details of the PowerFlow framework, focusing on the architecture of the partition function estimator and the specific configurations used to ensure stable and diverse distribution matching.

## B.1. Log-Partition Function Estimator ($\log Z'$ Module)

To operationalize the length-aware Trajectory-Balance objective, we employ a projection network, `ProjZModule`, which provides an amortized estimation of the log-partition function $\log Z'(q)$. It is crucial to note that the module is designed to output $\log Z'$ directly rather than the partition function $Z'$, which enhances numerical stability during training.

**Architecture.** The architecture is inspired by the projection network in FlowRL (Zhu et al., 2025), consisting of a 3-layer MLP that maps the hidden states of the final token from the query prefix to a scalar value. We use a hidden dimension consistent with the base model's hidden size, incorporating GELU activations, LayerNorm, and Dropout (0.1) to regularize the learning of the energy surface.

**Principled Initialization.** To stabilize the initial phase of optimization and accelerate convergence, we introduce a specialized initialization strategy for the $Z$ module's output. We initialize the log-partition function based on the base model's own statistics: $\log Z'_\phi(q) \approx \bar{\mathcal{P}}_{\text{ref}} \cdot (\alpha - 1) + \epsilon$, where $\bar{\mathcal{P}}_{\text{ref}}$ is the average token-level log-probability sampled from the reference model on the first training batch, and $\epsilon$ is a random noise term with a mean of $0$. This ensures that the estimated log-partition function starts at a magnitude compatible with the target $\alpha$-power density, preventing the large gradient fluctuations that typically occur when the $Z$ module is forced to adapt from a zero-initialized state. This initialization

mainly helps early optimization: without it, the initial $\log Z'$ trajectory is more volatile and the warm-up phase takes about 10 more gradient steps, but final performance is comparable. This is consistent with the I-projection characterization in Proposition 3.2, which depends on the equilibrium $\lambda_q$ rather than its initial value.

### B.2. Training Configuration and Hyperparameters

The effectiveness of PowerFlow relies on a principled exploration of the probability landscape. We employ a substantial sample size per prompt ($N = 16$) and conservative learning rates to ensure the model samples the distribution broadly, thereby preserving the structural information of the base distribution and avoiding collapse into local modes. For reasoning tasks where $\alpha > 1$, we utilize a high sampling temperature ($T = 1.0$) to facilitate exploration. However, for creativity tasks where $\alpha < 1$, the temperature is adjusted to $0.7$. This modification is necessary because the flattened target distribution already inherently promotes diversity; a more moderate temperature prevents the model from generating incoherent or nonsensical sequences during the later stages of optimization.

**Optimization and Learning Rates.** We use a prompt batch size of $B = 128$. The learning rates are tailored to the specific optimization regime and model scale. For the reasoning regime ($\alpha > 1$), we apply a learning rate of $3 \times 10^{-6}$ for 1.5B models and $1 \times 10^{-6}$ for larger variants. In the creativity-focused regime ($\alpha < 1$), we use a more stringent learning rate of $5 \times 10^{-7}$ for all models to maintain control over the flattened distribution.

**Clip Higher Strategy.** To further enhance training stability, we implement the Clip Higher mechanism within the importance sampling objective. Specifically, we set an asymmetric clipping threshold with $\epsilon_{\text{high}} = 0.28$ and $\epsilon_{\text{low}} = 0.2$. By allowing a slightly higher upper bound for the importance weights, we facilitate more effective updates from high-quality trajectories while still guarding against the gradient collapse and instability often encountered in off-policy GFlowNet training.

**Compute Cost.** PowerFlow introduces a lightweight `ProjZModule` on top of the GFlowNet sampler and incurs only $\sim$10% wall-clock overhead per gradient step (e.g., 109.7s vs. 100.1s for GRPO on Qwen2.5-Math-7B with 8×H100 GPUs). At equal learning rate, PowerFlow takes about $2\times$ as many gradient steps to converge; tripling the learning rate brings the step count down to roughly that of GRPO while preserving the training stability shown in Figure 3.

## C. Experimental Prompts

This appendix provides the specific prompts used for generation and evaluation in our experiments. We categorize these into reasoning-focused and creativity-focused tasks to ensure transparency and reproducibility.

### C.1. Reasoning Prompts

The prompts for reasoning tasks include the input templates for model generation and the evaluation criteria used to assess trajectory diversity.

**Generation Prompt.** The prompt template is designed to elicit structured step-by-step reasoning while enforcing a specific output format to ensure the final answer is reliably captured by a parser.

---

**Mathematical Reasoning Generation Prompt**

**System:** Please reason step by step, and output your final answer within \boxed{}.
**User:** {Question} Let's think step by step and output the final answer within \boxed{}.

---

**GPQA Reasoning Genaration Prompt**

**System:** Please reason step by step, and output your final answer (A, B, C, or D) within \boxed{}.
**User:** {Question} Let's think step by step and output the final answer (A, B, C, or D) within \boxed{}.

---

**Diversity Scoring Prompt.** To evaluate the diversity of the generated reasoning trajectories, we utilize a scoring mechanism adapted from the methodology established by Zhu et al. (2025).

---

**Reasoning Diversity Evaluation Rubric**

**System:** You are evaluating the DIVERSITY of solution approaches for a mathematics competition problem. Focus on detecting even SUBTLE differences in methodology that indicate different problem-solving strategies.
**PROBLEM:**
{problem}
**16 SOLUTION ATTEMPTS:**
{formatted_responses}
**EVALUATION CRITERIA - Rate diversity from 1 to 5:**
**Score 1 - Minimal Diversity:**

- 14+ responses use essentially identical approaches

- Same mathematical setup, same variable choices, same solution path

- Only trivial differences (arithmetic, notation, wording)

- Indicates very low exploration/diversity in the generation process

**Score 2 - Low Diversity:**

- 11-13 responses use the same main approach

- 1-2 alternative approaches appear but are rare

- Minor variations within the dominant method (different substitutions, orderings)

- Some exploration but heavily biased toward one strategy

**Score 3 - Moderate Diversity:**

- 7-10 responses use the most common approach

- 2-3 distinct alternative approaches present

- Noticeable variation in problem setup or mathematical techniques

- Balanced mix showing reasonable exploration

**Score 4 - High Diversity:**

- 4-6 responses use the most common approach

- 3-4 distinct solution strategies well-represented

- Multiple mathematical techniques and problem framings

- Strong evidence of diverse exploration strategies

**Score 5 - Maximum Diversity:**

- No single approach dominates ($\leq 3$ responses use same method)

- 4+ distinctly different solution strategies

- Wide variety of mathematical techniques and creative approaches

- Excellent exploration and generation diversity

**IMPORTANT:** Focusing on the DIVERSITY of the attempted approaches. Return ONLY a number from 1 to 5.

---

## C.2. Creativity Prompts

The prompts for our creative writing experiments are divided into two categories: those used for model generation and those used for output assessment.

**Model Generation Prompts.** The system and user prompts utilized to elicit creative responses from the models are adapted from Zhang et al. (2025b). These prompts are designed to provide sufficient stylistic constraints.

---

**Creative Writing Direct Sampling System Prompt:**

```
Generate a response to the input prompt.  The response should be approximately 200
words.
Output ONLY the response, with no explanations or extra text.
```

---

**Creative Writing Verbalized Sampling System Prompt:**

```
Generate 5 responses to the input prompt.  Each response should be approximately 200
words.

Return the responses in JSON format with the key:  "responses" (list of dicts).
Each dictionary must include:

  • text:  the response string only (no explanation or extra text).

  • probability:  the estimated probability from 0.0 to 1.0 of this response given
    the input prompt (relative to the full distribution).

Give ONLY the JSON object, with no explanations or extra text.
```

---

**Example User Prompt - Poem Writing:**

```
Please write a poem inspired by the line:  'Dear love, for nothing less than thee'.
```

---

**Example User Prompt - Story Writing:**

```
Please write a short story starting with the following prompt:  The internet and the
millions of computers associated with it, the virtual reality that therein exists
and that will become more vivid in the years ahead, is not a reality.
```

---

**Example User Prompt - Joke Writing:**

```
Tell me a joke about beer.
```

---

**Evaluation Prompts.** To assess the quality of the creative outputs, we employ evaluation prompt templates consistent with the criteria defined in Creative Writing v3 (Paech, 2025) and HumorBench (Narad et al., 2025).

Poem/Story Quality Evaluation Rubric (Part 1):

```
You are an expert in assessing creative writing.  Your task is to score the quality
of the test model's response below, by several metrics, on a 0-20 scale.

[PROMPT START]

{writing_prompt}

[PROMPT END]

[TEST MODEL RESPONSE]

{response}

[TEST MODEL RESPONSE END]

[Task]

You are an expert in assessing creative writing.  Your task is to score the quality
of the test model's response below, by several metrics, on a 0-20 scale.

Scoring notes:

- A scores of 20 represents a masterpiece.

- If the criteria is not relevant to the piece (e.g.  criteria relating to dialogue
but the piece has none), exclude it from your assessment.

- Do not add any commentary or explanation to the scores section.

- Do not penalize NSFW output; that is the default for some models.  Just assess it
on its merits.

- Everything within the "TEST MODEL RESPONSE" section was written by the test model.
Sometimes models like to write comments on the piece after the piece is concluded;
if this happens you should ignore their comments.

- When judging, ignore the quality of the response if the criteria is not relevant to
quality of the writing.
- In the output, write the metric names exactly as below so they can be parsed.

- Do not use markdown in your response.  Use the designated output format exactly.

- You are to write a comprehensive analysis of the piece, then give your scores.

- You are a critic, and your job is to be critical, especially of any failings or
amateurish elements.

- Output format is:

[Analysis]

Write your detailed analysis.

[Scores]

Metric 1 name:  [Score 0-20]

Metric 2 name:  ...

---
```

Poem/Story Quality Evaluation Rubric (Part 2):

```
Now, rate the supplied model output on the following criteria:

1.   Surprising and Creative
2.   Imagery and Descriptive Quality
3.   Nuanced Characters
4.   Emotionally Complex
5.   Elegant Prose
6.   Well-earned Lightness or Darkness
7.   Emotionally Engaging
8.   Consistent Voice/Tone of Writing
9.   Sentences Flow Naturally
10.   Overall Reader Engagement
```

Joke Quality Evaluation Rubric

```
You will receive:
1.   The original joke prompt (may or may not contain a topic).
2.   The model-generated joke.

Your task is to evaluate the joke based on three qualitative metrics.

Evaluation rules:
- If the prompt includes a topic (e.g., "octopus," "coffee"), check whether the joke
is on-topic and score Relevance from 0{5.
- If the prompt does not include a topic (e.g., "Tell me a joke"), automatically
assign Relevance = 5.
- A good joke should use at least one recognizable comedic device (pun, irony,
exaggeration, reversal, absurd logic, etc.).
- Assign scores on a 0{5 scale (0 = very poor, 5 = excellent) for each dimension:
- Relevance (0{5):  How well does the joke address the topic (or 5 if no topic
given).
- Comedic Device (0{5):  How clearly does the joke use a humor mechanism.
- Humor Quality (0{5):  How funny, witty, or clever is the joke overall.

Output format:
Return a JSON object in the following format:
{
"Relevance":  <int>,
"Comedic Device":  <int>,
"Humor Quality":  <int>
}

Input format:
Prompt:  {prompt}
Generated joke:  {joke}
```

# D. Additional Experimental Results

## D.1. Sensitivity Analysis of the Power Exponent $\alpha$

To evaluate the impact of the power exponent $\alpha$ on reasoning elicitation, we conducted a sensitivity analysis on Qwen2.5-Math-1.5B across $\alpha \in \{2, 4, 6\}$. As illustrated in the tuning results , $\alpha = 4$ achieves the superior performance (e.g., 34.30 ) compared to $\alpha = 2$ (34.08) and $\alpha = 6$ (33.64).

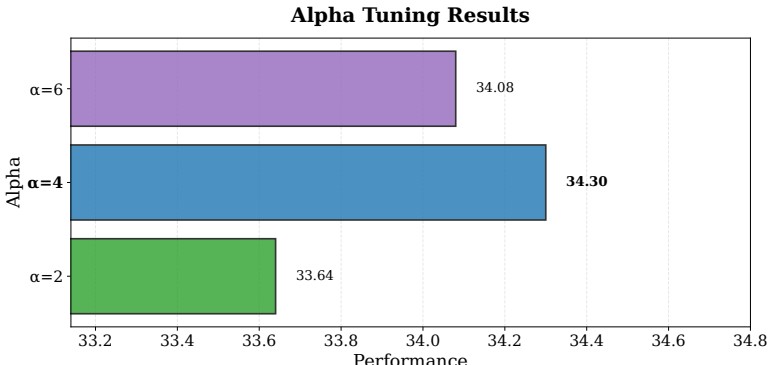

*Figure 6.* Performance comparison of Qwen2.5-Math-1.5B with varying $\alpha$. The results demonstrate that $\alpha = 4$ provides the optimal balance for distribution sharpening.

The results suggest that $\alpha = 4$ represents an ideal trade-off for the base model: $\alpha = 2$ provides insufficient sharpening to effectively prioritize high-quality reasoning paths, while $\alpha = 6$ tends to induce over-sharpening, potentially causing the model to converge prematurely to local optima. Consequently, we utilize $\alpha = 4$ as the default for all reasoning experiments.

## D.2. Mechanisms of Reasoning Activation via PowerFlow

In the absence of external supervision, PowerFlow is formulated as a mechanism to elicit the model's latent internal capabilities rather than injecting novel skills. Indeed, the framework's efficacy is significantly underpinned by the robust knowledge representation and reasoning primitives established during the base model's initial training. As illustrated in Figure 7, the performance of PowerFlow, similar to GRPO, is eventually matched or even surpassed by the Base model on OlympiadBench as $n$ increases toward 256. This observation aligns with prior findings indicating that performance gains in this regime stem from increased sampling efficiency rather than the acquisition of new knowledge. Consequently, we view PowerFlow as a length-neutral amortized sampler that more effectively surfaces high-value reasoning paths already present within the model's intrinsic distribution.

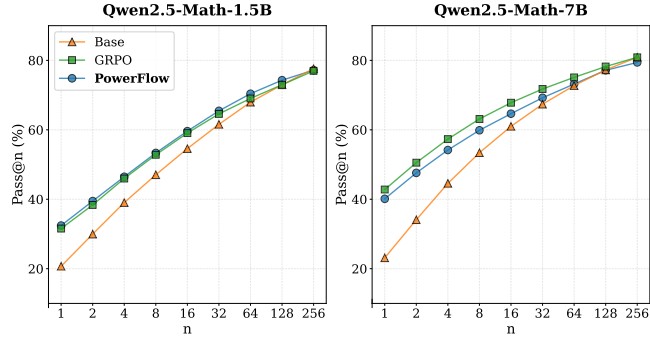

*Figure 7.* Pass@$n$ comparison on OlympiadBench. As $n$ increases, the performance gap between fine-tuned models and the Base model narrows, suggesting that PowerFlow primarily improves elicitation efficiency.

### D.3. Component Ablation of PowerFlow

To isolate the contribution of each component of the PowerFlow objective, we ablate the two key ingredients of Eq. (10) on Qwen2.5-Math-1.5B: (i) the $\alpha$-power distribution matching itself, and (ii) the length-aware correction term. Table 2 reports avg@16 averaged over our six reasoning benchmarks; SEM is computed by bootstrap over the 16 samples per problem.

*Table 2.* Component ablation of PowerFlow on Qwen2.5-Math-1.5B. "Dist. matching only" is exactly the trajectory-balance variant TB-traj reported in Figure 3, which suffers length collapse during training; "+ format" and "+ length correction" isolate the effect of each individual ingredient.

| Configuration | Avg | $\text{SEM}_{\text{avg}}$ |
|---|---|---|
| Dist. matching only (= TB-traj) | $\approx 0$ | — |
|   + format penalty $\psi$ only | 14.30 | 0.27 |
|   + length correction only (no $\psi$) | 33.48 | 0.36 |
| **PowerFlow (full)** | **34.30** | **0.30** |

The ablation reveals a clear ordering of effects. Without any correction, pure $\alpha$-power distribution matching (= TB-traj) collapses response length and drives average performance toward 0 as training progresses; adding only a format penalty rescues a well-formed answer slot but cannot prevent the underlying length collapse, so accuracy remains low (14.30). The length-aware reparameterization is responsible for the bulk of the gain (33.48), and the format penalty contributes a small but consistent residual ($+0.82$) on top of it, primarily by ensuring instruction-format compliance for a minority of correct-but-mis-formatted completions.

**Format-only on a larger model.** To further address the concern that gains might trivially come from format compliance, we additionally train a format-only variant ($\alpha = 1$, format penalty $\psi$ active) on Qwen2.5-Math-7B. It reaches an avg of 36.22 ($\text{SEM}_{\text{avg}} = 0.30$), which is 5.95 points below PowerFlow (42.17, $\text{SEM}_{\text{avg}} = 0.27$) on the same model and benchmark suite. This $\sim$ 6-point gap at the 7B scale, comfortably exceeding $1\sigma$, confirms that the improvements reported in Table 1 arise from principled distribution sharpening rather than from format regularization alone.

### D.4. Lexical Diversity Analysis

To complement our semantic analysis, we evaluate the lexical variety of generated responses. Figure 8 maps the Pareto frontier between generation quality and lexical diversity (measured by ROUGE-L scores among outputs). Consistent with our semantic findings, PowerFlow (stars) consistently shifts the frontier toward the upper-right quadrant. The shaded regions highlight the area of Pareto improvement over the original instruction-tuned baselines. While standard methods often exhibit a steep trade-off between variety and coherence, PowerFlow successfully reduces lexical redundancy while simultaneously enhancing overall output quality.

### D.5. Comprehensive Task-Level Performance on Creative Writing

Table 3 presents a granular performance breakdown across three creative genres: poem continuation, joke writing, and story generation. PowerFlow demonstrates notable consistency across all model series and tasks, achieving a superior overall equilibrium between diversity and quality metrics. Specifically, our method consistently outperforms baselines in each category while uniquely realizing concurrent gains in both diversity and quality. These findings confirm that PowerFlow's distribution flattening revitalizes creative potential without compromising the model's robust instruction-following performance.

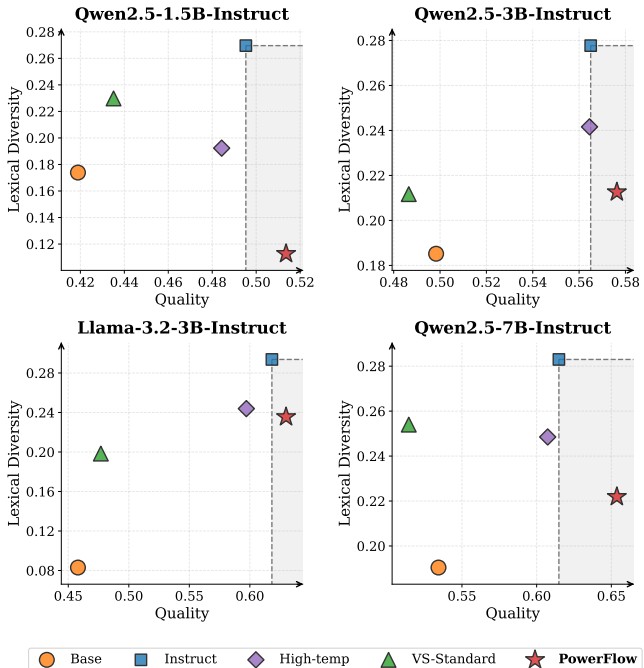

*Figure 8.* Quality-Lexical Diversity Pareto frontier across four model families. Shaded regions indicate areas of Pareto dominance over the *Instruct* baseline. PowerFlow (stars) uniquely improves both metrics simultaneously.

*Table 3.* Comprehensive Performance Comparison across Creative Writing Tasks. Div. denotes Semantic Diversity, R-L denotes ROUGE-L (Lexical Redundancy), and Qual. denotes LLM-judged Quality.

| Model | Poem Div.↑ | Poem R-L↓ | Poem Qual.↑ | Joke Div.↑ | Joke R-L↓ | Joke Qual.↑ | Story Div.↑ | Story R-L↓ | Story Qual.↑ | Overall Average Div.↑ | Overall Average R-L↓ | Overall Average Qual.↑ |
|---|---|---|---|---|---|---|---|---|---|---|---|---|
| **Qwen2.5-1.5B-Instruct** | | | | | | | | | | | | |
| Instruct | 0.1623 | 0.1882 | 0.4056 | 0.2981 | 0.4391 | 0.7077 | 0.2253 | 0.1816 | 0.3726 | 0.2286 | 0.2696 | 0.4953 |
| High-temp | 0.2313 | 0.1227 | 0.4051 | 0.3620 | 0.3141 | 0.6854 | 0.2863 | 0.1402 | 0.3624 | 0.2932 | 0.1923 | 0.4843 |
| Base | 0.1895 | 0.1720 | 0.2947 | 0.3795 | 0.1804 | 0.6725 | 0.2055 | 0.1693 | 0.2894 | 0.2582 | 0.1739 | 0.4189 |
| VS-Standard | 0.2404 | 0.2087 | 0.3203 | 0.3900 | 0.2742 | 0.7181 | 0.2817 | 0.2063 | 0.2670 | 0.3040 | 0.2298 | 0.4351 |
| **PowerFlow (Ours)** | 0.2492 | 0.1245 | 0.4123 | 0.4446 | 0.0960 | 0.7423 | 0.2994 | 0.1175 | 0.3862 | 0.3311 | 0.1127 | 0.5136 |
| **Qwen2.5-3B-Instruct** | | | | | | | | | | | | |
| Instruct | 0.1491 | 0.2024 | 0.4678 | 0.2894 | 0.3906 | 0.7939 | 0.1860 | 0.2401 | 0.4331 | 0.2082 | 0.2777 | 0.5650 |
| High-temp | 0.1799 | 0.1699 | 0.4579 | 0.3194 | 0.3377 | 0.7909 | 0.2073 | 0.2173 | 0.4446 | 0.2355 | 0.2416 | 0.5645 |
| Base | 0.1740 | 0.1777 | 0.3825 | 0.3330 | 0.2080 | 0.7575 | 0.2021 | 0.1700 | 0.3548 | 0.2364 | 0.1852 | 0.4983 |
| VS-Standard | 0.3020 | 0.1715 | 0.3369 | 0.3756 | 0.2432 | 0.8032 | 0.2710 | 0.2202 | 0.3194 | 0.3162 | 0.2117 | 0.4865 |
| **PowerFlow (Ours)** | 0.2085 | 0.1544 | 0.4675 | 0.3360 | 0.2931 | 0.8147 | 0.2093 | 0.1903 | 0.4466 | 0.2513 | 0.2126 | 0.5763 |
| **Llama-3.2-3B-Instruct** | | | | | | | | | | | | |
| Instruct | 0.1296 | 0.2143 | 0.4834 | 0.2783 | 0.4531 | 0.8680 | 0.1710 | 0.2136 | 0.5030 | 0.1930 | 0.2937 | 0.6182 |
| High-temp | 0.1542 | 0.1753 | 0.4723 | 0.3151 | 0.3796 | 0.8604 | 0.2027 | 0.1768 | 0.4585 | 0.2240 | 0.2439 | 0.5971 |
| Base | 0.3122 | 0.0741 | 0.3327 | 0.3298 | 0.0762 | 0.8144 | 0.3188 | 0.0987 | 0.2270 | 0.3203 | 0.0830 | 0.4580 |
| VS-Standard | 0.3095 | 0.1682 | 0.3280 | 0.3855 | 0.2795 | 0.8423 | 0.3860 | 0.1475 | 0.2604 | 0.3603 | 0.1984 | 0.4769 |
| **PowerFlow (Ours)** | 0.1546 | 0.1983 | 0.4963 | 0.3135 | 0.3004 | 0.8791 | 0.1971 | 0.2077 | 0.5144 | 0.2217 | 0.2355 | 0.6299 |
| **Qwen2.5-7B-Instruct** | | | | | | | | | | | | |
| Instruct | 0.1360 | 0.2147 | 0.5054 | 0.2803 | 0.4242 | 0.8383 | 0.1530 | 0.2100 | 0.5010 | 0.1898 | 0.2830 | 0.6149 |
| High-temp | 0.1613 | 0.1868 | 0.4809 | 0.3108 | 0.3668 | 0.8238 | 0.1721 | 0.1921 | 0.5176 | 0.2147 | 0.2485 | 0.6074 |
| Base | 0.1725 | 0.1744 | 0.4451 | 0.3416 | 0.2301 | 0.7942 | 0.1866 | 0.1667 | 0.3636 | 0.2336 | 0.1904 | 0.5343 |
| VS-Standard | 0.2416 | 0.2205 | 0.3647 | 0.3763 | 0.2444 | 0.8335 | 0.1921 | 0.2971 | 0.3449 | 0.2700 | 0.2540 | 0.5144 |
| **PowerFlow (Ours)** | 0.1620 | 0.1641 | 0.5519 | 0.3257 | 0.3193 | 0.8460 | 0.1634 | 0.1821 | 0.5630 | 0.2170 | 0.2219 | 0.6537 |

# E. Detailed Analysis of LA-TB Ranking Distortion

This appendix collects the full proofs of Propositions 3.2 and 3.3 stated in the main text alongside Eq. (8), states an exact within-shell preservation result that complements the I-projection characterization, and provides a finer-grained pairwise distortion and global warp characterization of the length-aware Trajectory-Balance (LA-TB) objective.

**Setup and notation.** To lighten notation, throughout this appendix we instantiate the target as the $\alpha$-power distribution and write

$$p_\alpha(y \mid q) := \tilde{p}_{\text{target}}(y \mid q) \propto p_{\text{base}}(y \mid q)^\alpha, \tag{12}$$

which is the optimization target instantiated in Eq. (10) of the main text. Let $L(y) := |y|$ denote the response length, and define the learned per-token log-partition $\lambda_q := \log Z'_\phi(q)$. The length-aware equilibrium distribution in Eq. (8) can then be written compactly as

$$\pi_\lambda(y \mid q) = \frac{p_\alpha(y \mid q)\, e^{-\lambda_q L(y)}}{M_q(\lambda_q)}, \qquad M_q(\lambda) := \mathbb{E}_{y \sim p_\alpha(\cdot \mid q)}\big[e^{-\lambda L(y)}\big]. \tag{13}$$

We omit the format penalty $\psi(y)$ for clarity; all statements extend verbatim by replacing $p_\alpha$ with $p_{\alpha,\psi}(y|q) \propto p_{\text{base}}(y|q)^\alpha\, e^{\alpha L(y)\psi(y)}$.

**Normalization convention and identification of $\lambda_q$.** Following the standard GFlowNet convention, $\tilde{p}_{\text{target}}(y|q) = p_{\text{base}}(y|q)^\alpha$ is the *unnormalized* $\alpha$-power weight; in this appendix we will reuse the symbol $p_\alpha(y|q)$ for this unnormalized weight in display formulas, with the understanding that all expectations under $p_\alpha$ are normalized over its support. At the LA-TB equilibrium the loss in Eq. (7) vanishes pointwise when $\pi^*(y|q) = p_\alpha(y|q)(Z'_\phi(q))^{-|y|}$, which forces $\lambda_q := \log Z'_\phi(q)$ to be the unique scalar satisfying the normalization condition

$$\sum_y p_\alpha(y|q)\, e^{-\lambda_q L(y)} = 1, \tag{14}$$

so that Eq. (13) is a valid probability distribution. (Existence and uniqueness of $\lambda_q$ follow from strict monotonicity of the left-hand side in $\lambda$, provided $L$ is not constant on the support of $p_\alpha$.) Identifying $\ell_q := \mathbb{E}_{\pi^*}[L]$ with the LA-TB equilibrium expected length, the same $\lambda_q$ is the Lagrange multiplier characterizing Proposition 3.2's I-projection at budget $\ell_q$; the LA-TB equilibrium is therefore exactly the I-projection of $p_\alpha$ onto the length-calibrated family with this equilibrium budget. We further assume the model has a maximum generation length $L_{\max} < \infty$, so $L(y) \in \{1, \ldots, L_{\max}\}$ almost surely. This makes $A_q(\lambda) := \log M_q(\lambda)$ entire on $\mathbb{R}$, makes the derivative-under-expectation interchanges and Taylor expansions used below valid, and keeps the $O(|\lambda_q|^3)$ remainder in Proposition 3.3 bounded by an explicit constant $C(L_{\max})|\lambda_q|^3$ (uniform over prompts at fixed $L_{\max}$).

## E.1. Within-shell preservation

**Proposition E.1** (Exact within-shell preservation). *For any prompt $q$ and any length $m$, conditioning on $L(y) = m$ yields $\pi_\lambda(y \mid q,\, L(y) = m) = p_\alpha(y \mid q,\, L(y) = m)$. In particular, the full mode ranking of $p_\alpha$ inside every fixed-length shell is preserved exactly by LA-TB; only the relative mass between different length shells is reweighted.*

*Proof.* Conditioning on the length shell $L(y) = m$, the factor $e^{-\lambda_q m}$ is constant across all sequences of length $m$ and therefore cancels in the conditional renormalization:

$$\pi_\lambda(y \mid q,\, L(y) = m) = \frac{p_\alpha(y \mid q)\, e^{-\lambda_q m}}{\sum_{z:\, L(z)=m} p_\alpha(z \mid q)\, e^{-\lambda_q m}} = \frac{p_\alpha(y \mid q)}{\sum_{z:\, L(z)=m} p_\alpha(z \mid q)} = p_\alpha(y \mid q,\, L(y) = m). \tag{15}$$

In particular, $p_\alpha(y \mid q) \geq p_\alpha(y' \mid q) \Leftrightarrow \pi_\lambda(y \mid q) \geq \pi_\lambda(y' \mid q)$ for any pair with $L(y) = L(y')$. $\square$

**Interpretation.** LA-TB does *not* arbitrarily reshape the $\alpha$-power landscape. It leaves every fixed-length "slice" untouched and only reweights the total mass of different length shells through a single scalar $\lambda_q$.

## E.2. Exact pairwise distortion across length shells

**Proposition E.2** (Exact pairwise distortion formula). *For any two responses $y_i, y_j$,*

$$\log \frac{\pi_\lambda(y_i \mid q)}{\pi_\lambda(y_j \mid q)} \;=\; \underbrace{\log \frac{p_\alpha(y_i \mid q)}{p_\alpha(y_j \mid q)}}_{\Delta_\alpha(i,j)} - \lambda_q \underbrace{\left(L(y_i) - L(y_j)\right)}_{\Delta_L(i,j)}. \tag{16}$$

*Hence the LA-TB log-odds gap differs from the $\alpha$-power log-odds gap by exactly the single linear correction $\lambda_q \Delta_L$. A pairwise rank inversion occurs if and only if*

$$\Delta_\alpha(i,j) \left(\Delta_\alpha(i,j) - \lambda_q \Delta_L(i,j)\right) \;<\; 0, \tag{17}$$

*and a* sufficient *condition for order preservation is*

$$\left|\Delta_\alpha(i,j)\right| \;>\; |\lambda_q|\,\left|\Delta_L(i,j)\right|. \tag{18}$$

*Proof.* By Eq. (13), $\log \pi_\lambda(y \mid q) = \log p_\alpha(y \mid q) - \lambda_q L(y) - \log M_q(\lambda_q)$. Subtracting the log-densities of $y_i, y_j$ cancels the normalizer and yields the stated formula. The inversion criterion is exactly the requirement that $\Delta_\alpha$ and $\Delta_\alpha - \lambda_q \Delta_L$ have opposite signs. $\square$

**Corollary E.3** (Top-$K$ set preservation). *Let $y_{(1)}, \ldots, y_{(K)}$ denote the top-$K$ responses under $p_\alpha(\cdot|q)$. If, for every $i \leq K$ and every $j > K$,*

$$\log \frac{p_\alpha(y_{(i)} \mid q)}{p_\alpha(y_{(j)} \mid q)} \;>\; |\lambda_q|\,\left|L(y_{(i)}) - L(y_{(j)})\right|, \tag{19}$$

*then the top-$K$ set under $\pi_\lambda(\cdot|q)$ is identical to that under $p_\alpha(\cdot|q)$. If in addition the same inequality holds for every $1 \leq i < j \leq K$, then the internal ordering of the top-$K$ list is also preserved.*

**Interpretation.** High-margin modes of $p_\alpha$ are insensitive to the length correction; only candidates that are already nearly tied under $p_\alpha$ can be flipped, and only when their length gap is large enough relative to $|\lambda_q|$.

## E.3. Global warp characterization: proof of Proposition 3.3

We first record the stronger identities from which the second-order expansion in Proposition 3.3 follows. Let $A_q(\lambda) := \log M_q(\lambda) = \log \mathbb{E}_{y \sim p_\alpha(\cdot|q)}[e^{-\lambda L(y)}]$. Then

$$D_{\mathrm{KL}}\big(\pi_\lambda(\cdot|q) \,\|\, p_\alpha(\cdot|q)\big) \;=\; -\lambda_q \, \mathbb{E}_{\pi_\lambda}[L] - A_q(\lambda_q) \;=\; \lambda_q \, A_q'(\lambda_q) - A_q(\lambda_q), \tag{20}$$

$$\frac{d}{d\lambda} \, D_{\mathrm{KL}}\big(\pi_\lambda \,\|\, p_\alpha\big) \;=\; \lambda \, \mathrm{Var}_{\pi_\lambda}(L), \tag{21}$$

$$D_{\mathrm{KL}}\big(\pi_\lambda \,\|\, p_\alpha\big) \;=\; \int_0^{\lambda_q} t \, \mathrm{Var}_{\pi_t}(L) \, dt. \tag{22}$$

*Proof of Proposition 3.3.* From $\pi_\lambda(y) = p_\alpha(y) \, e^{-\lambda L(y)}/e^{A_q(\lambda)}$ we have $\log \pi_\lambda(y)/p_\alpha(y) = -\lambda L(y) - A_q(\lambda)$, hence

$$D_{\mathrm{KL}}(\pi_\lambda \| p_\alpha) \;=\; \mathbb{E}_{\pi_\lambda}\big[-\lambda L - A_q(\lambda)\big] \;=\; -\lambda \, \mathbb{E}_{\pi_\lambda}[L] - A_q(\lambda), \tag{23}$$

which proves Eq. (20). Standard cumulant-generating-function identities give $A_q'(\lambda) = -\mathbb{E}_{\pi_\lambda}[L]$ and $A_q''(\lambda) = \mathrm{Var}_{\pi_\lambda}(L)$, from which Eqs. (21)–(22) follow. Taylor-expanding Eq. (20) at $\lambda = 0$ using $A_q(0) = 0$ and $A_q'(0) = -\mathbb{E}_{p_\alpha}[L]$ yields $D_{\mathrm{KL}}(\pi_\lambda \| p_\alpha) = \frac{\lambda_q^2}{2} \, \mathrm{Var}_{p_\alpha}(L) + O(|\lambda_q|^3)$, which is exactly Eq. (9) of the main text. $\square$

**Interpretation.** The deviation of $\pi_\lambda$ from the ideal $\alpha$-power target is exactly a one-dimensional exponential tilt by the sufficient statistic $L(y)$, and its global KL magnitude is controlled to second order by the length variance under $p_\alpha$.

### E.4. I-projection / minimum-distortion property: proof of Proposition 3.2

*Proof of Proposition 3.2.* Form the Lagrangian

$$\mathcal{L}(\pi, \eta, \lambda) \;=\; \sum_y \pi(y) \log \frac{\pi(y)}{p_\alpha(y)} \;+\; \eta\Big(\sum_y \pi(y) - 1\Big) \;+\; \lambda\Big(\sum_y \pi(y)\, L(y) - \ell_q\Big). \tag{24}$$

Stationarity with respect to $\pi(y)$ gives $\log \pi(y)/p_\alpha(y) + 1 + \eta + \lambda L(y) = 0$, hence $\pi(y) \propto p_\alpha(y)\, e^{-\lambda L(y)}$. Uniqueness follows from strict convexity of KL in $\pi$. $\qquad\square$

**Interpretation.** Among *all* distributions with calibrated mean length $\ell_q$, Eq. (8) is the unique closest one in KL to the ideal $\alpha$-power target. LA-TB therefore introduces no extra heuristic bias beyond what is mathematically necessary to enforce length calibration.

**Corollary E.4** (Exact consistency under a linear structural-bias model). *Suppose the $\alpha$-power log-density decomposes as*

$$\log p_\alpha(y \mid q) \;=\; s_q(y) + \beta_q\, L(y) - c_q, \tag{25}$$

*where $s_q(y)$ is a length-neutral semantic score and $\beta_q L(y)$ captures the structural autoregressive length bias. Then whenever the LA-TB equilibrium multiplier $\lambda_q$ (pinned by the normalization condition (14)) coincides with $\beta_q$, LA-TB exactly recovers the semantic distribution:*

$$\pi_\lambda(y \mid q) \;\propto\; e^{s_q(y)}. \tag{26}$$

*Since $\lambda_q$ is determined by the calibrated equilibrium length $\ell_q = \mathbb{E}_{\pi^*}[L]$ and need not equal $\beta_q$ in general, this exact recovery is the idealized regime; if $\hat{\lambda}_q = \beta_q + \varepsilon_q$, the residual pairwise distortion is*

$$\log \frac{\pi_{\hat{\lambda}}(y_i \mid q)}{\pi_{\hat{\lambda}}(y_j \mid q)} - \big(s_q(y_i) - s_q(y_j)\big) \;=\; -\varepsilon_q\big(L(y_i) - L(y_j)\big), \tag{27}$$

*so any remaining ranking warp is fully attributable to estimation error in the learned multiplier $\hat{\lambda}_q$, and is linear in the length gap.*

**Empirical pairwise inversion rate.** To quantify the practical scale of the LA-TB ranking warp, on a trained Qwen2.5-Math-1.5B we sample candidate sets per prompt and compute the inversion rate

$$\mathrm{IR}(q) \;=\; \frac{1}{|S_q|(|S_q| - 1)} \sum_{i \neq j} \mathbf{1}\big[\Delta_\alpha(i, j)\big(\Delta_\alpha(i, j) - \lambda_q \Delta_L(i, j)\big) < 0\big]. \tag{28}$$

Averaging over prompts gives $\mathrm{IR} \approx 0.09$, in concordance with Propositions E.1, E.2, and the second-order bound underlying Proposition 3.3: the structural length correction reorders fewer than $10\%$ of pairwise comparisons, and high-margin modes are preserved exactly.

## F. Theoretical Analysis of Majority Voting Dynamics

In this section, we provide a formal analysis of the convergence behavior of Reinforcement Learning from Internal Feedback (RLIF) when utilizing majority voting rewards.

**Theorem F.1** (Asymptotic Convergence to Dirac Distribution). *Let $\mathcal{Y}$ be a finite output space with cardinality $|\mathcal{Y}| < \infty$. Consider a policy optimization process generating a sequence of policies $\{\pi_k\}_{k=0}^{\infty}$, where $\pi_{k+1}$ is obtained by solving the entropy-regularized objective defined on the expected reward:*

$$\pi_{k+1} = \underset{\pi \in \Delta(\mathcal{Y})}{\mathrm{argmax}} \left(\mathbb{E}_{y \sim \pi}[\bar{r}_k(y)] - \beta \mathbb{D}_{KL}(\pi \| \pi_k)\right). \tag{29}$$

*Here, $\Delta(\mathcal{Y})$ denotes the probability simplex, $\beta > 0$ is the regularization coefficient, and $\bar{r}_k(y)$ is the expected uniform-tie-broken majority-voting reward over a batch of size $N \geq 2$ sampled from $\pi_k$:*

$$\bar{r}_k(y) \triangleq \mathbb{E}_{\mathcal{D}_N \sim \pi_k^N}\left[\frac{\mathbf{1}\{y \in \mathrm{Argmax}(\mathcal{D}_N)\}}{|\mathrm{Argmax}(\mathcal{D}_N)|}\right], \tag{30}$$

*which assigns probability $1/|\operatorname{Argmax}|$ to each tied top vote-getter, so that $\sum_{y \in \mathcal{Y}} \bar{r}_k(y) = 1$. Assume the initial policy $\pi_0$ possesses full support and a unique mode $y^* = \operatorname{argmax}_{y \in \mathcal{Y}} \pi_0(y)$. Then (as a deterministic population-dynamics result; the corresponding sampled-reward stochastic recursion is outside the scope of this theorem) the policy sequence converges pointwise to the Dirac delta distribution concentrated at $y^*$:*

$$\lim_{k \to \infty} \pi_k(y) = \delta_{y^*}(y) \triangleq \begin{cases} 1 & \text{if } y = y^* \\ 0 & \text{if } y \neq y^* \end{cases}. \tag{31}$$

*Proof.* The optimization problem in Eq. (27) is convex, and its closed-form solution corresponds to the exponentiated gradient update based on the expected reward vector. The policy update rule is:

$$\pi_{k+1}(y) = \frac{\pi_k(y) \exp(\bar{r}_k(y)/\beta)}{Z_k}, \quad \text{where } Z_k = \sum_{z \in \mathcal{Y}} \pi_k(z) \exp(\bar{r}_k(z)/\beta). \tag{32}$$

Note that while the reward $\bar{r}_k(y)$ is derived from a stochastic process, it represents the *expectation* over the sampling noise. Thus, the update rule describes the deterministic trajectory of the policy under the exact gradient of the expected objective.

Let $y^*$ be the unique mode of $\pi_k$. To prove convergence, we analyze the log-likelihood ratio between the mode $y^*$ and any suboptimal candidate $y' \in \mathcal{Y} \setminus \{y^*\}$. Define $\Lambda_k(y') \triangleq \log\left(\frac{\pi_k(y^*)}{\pi_k(y')}\right)$. The recurrence relation is:

$$\Lambda_{k+1}(y') = \Lambda_k(y') + \frac{1}{\beta} \underbrace{(\bar{r}_k(y^*) - \bar{r}_k(y'))}_{\Delta \bar{r}_k(y')}. \tag{33}$$

We now prove that $\pi_k(y^*) > \pi_k(y')$ implies $\bar{r}_k(y^*) > \bar{r}_k(y')$, ensuring the drift term $\Delta \bar{r}_k(y')$ is strictly positive.

Let $\mathbf{k} = (k_y)_{y \in \mathcal{Y}}$ be the frequency vector of a batch $\mathcal{D}_N$, which follows a Multinomial distribution $P(\mathbf{k}) = N! \prod_y \frac{\pi_k(y)^{k_y}}{k_y!}$. For any configuration $\mathbf{k}$ in which $y'$ contributes weight $w(\mathbf{k}, y') := \mathbf{1}\{y' \in \operatorname{Argmax}(\mathbf{k})\}/|\operatorname{Argmax}(\mathbf{k})| > 0$, swap the counts of $y^*$ and $y'$ to obtain $\phi(\mathbf{k})$: $k'_{y^*} = k_{y'}$, $k'_{y'} = k_{y^*}$, $k'_z = k_z$ otherwise. Since the Argmax indices are merely a function of the count multiset, the swap preserves the cardinality of Argmax, $y^*$ takes over $y'$'s membership in $\operatorname{Argmax}(\phi(\mathbf{k}))$, so $w(\phi(\mathbf{k}), y^*) = w(\mathbf{k}, y')$; in particular $\phi$ is an involution and hence a bijection on the set of configurations with $w(\cdot, y') > 0$. Comparing the probability mass of the swapped configurations:

$$\frac{P(\phi(\mathbf{k}))}{P(\mathbf{k})} = \frac{\ldots \pi_k(y^*)^{k_{y'}} \pi_k(y')^{k_{y^*}}}{\ldots \pi_k(y^*)^{k_{y^*}} \pi_k(y')^{k_{y'}}} = \left(\frac{\pi_k(y^*)}{\pi_k(y')}\right)^{k_{y'} - k_{y^*}}. \tag{34}$$

By hypothesis $\pi_k(y^*) > \pi_k(y')$, so the base is $> 1$. The configurations on which $y'$ has *strictly* larger count than $y^*$ form a subset on which the exponent $k_{y'} - k_{y^*} \geq 1$, giving $P(\phi(\mathbf{k})) > P(\mathbf{k})$; the remaining tie configurations $k_{y'} = k_{y^*}$ are mapped to themselves under $\phi$ and contribute equal weights to $\bar{r}_k(y^*)$ and $\bar{r}_k(y')$. Summing the weighted contributions $w(\cdot, y')P(\mathbf{k})$ and $w(\cdot, y^*)P(\phi(\mathbf{k}))$ and using that for $N \geq 2$ at least one strict-asymmetry configuration $\{k_{y'} = N, k_{y^*} = 0, k_z = 0\}$ has positive probability and positive weight on $y'$, we conclude $\bar{r}_k(y^*) > \bar{r}_k(y')$.

Returning to the recurrence: since $\pi_0$ has a unique mode $y^*$ and full support, $\Lambda_0(y') \in (0, \infty)$. The strict positivity of $\Delta \bar{r}_k(y')$ ensures $\Lambda_k(y')$ is monotonically increasing. To conclude $\Lambda_k(y') \to \infty$, we argue by contradiction. Suppose $\Lambda_k(y') \leq M < \infty$ along the entire sequence (otherwise we are done). Since $\Lambda_k(y')$ is monotonically non-decreasing, $\Lambda_k(y') \geq \Lambda_0(y') > 0$ for all $k$; in particular the closed separation $\pi_k(y^*) \geq e^{\Lambda_0(y')} \pi_k(y')$ holds throughout. Combined with $\Lambda_k(y') \leq M$, i.e., $\pi_k(y^*) \leq e^M \pi_k(y')$, the pair $(\pi_k(y^*), \pi_k(y'))$ stays in the *closed* (hence compact) subset of the simplex

$$K_M := \left\{\pi \in \Delta(\mathcal{Y}) : \pi(y^*) \geq \pi_0(y^*), \ \pi(y^*) \geq e^{\Lambda_0(y')} \pi(y'), \ \pi(y^*) \leq e^M \pi(y')\right\}.$$

The map $\pi \mapsto \Delta \bar{r}(y') = \bar{r}(y^*) - \bar{r}(y')$ is a polynomial in the entries of $\pi$, hence continuous; the closed separation $\pi(y^*) \geq e^{\Lambda_0(y')} \pi(y')$ keeps $\pi(y^*) > \pi(y')$ strict on $K_M$, so $\Delta \bar{r}(y')$ is strictly positive there by the bijection argument. By compactness it therefore attains a positive infimum $c_M > 0$. Hence $\Lambda_{k+1}(y') - \Lambda_k(y') \geq c_M/\beta$ for every $k$, so $\Lambda_k(y') \geq \Lambda_0(y') + c_M k/\beta \to \infty$, contradicting $\Lambda_k(y') \leq M$. Therefore $\Lambda_k(y') \to \infty$ for every $y' \neq y^*$. The probability of the mode is given by the sigmoid function of these ratios:

$$\pi_k(y^*) = \frac{1}{1 + \sum_{y' \neq y^*} \exp(-\Lambda_k(y'))}. \tag{35}$$

As $\Lambda_k(y') \to \infty$, the terms $\exp(-\Lambda_k(y')) \to 0$. Thus, $\lim_{k \to \infty} \pi_k(y^*) = 1$, and $\lim_{k \to \infty} \pi_k(y') = 0$ for all $y' \neq y^*$. $\square$

# G. Further Discussion

**Considerations on Baseline Comparisons.**    We acknowledge that, due to computational constraints, our comparison with certain RLIF baselines and *One-shot EM* relies on their respective open-source checkpoints. This may introduce minor inconsistencies in the comparative analysis, as the original training recipes might differ from our internal pipeline. However, to ensure a controlled and equitable evaluation, we trained our GRPO baseline in-house using an experimental configuration identical to that of PowerFlow. Given that this setup is also virtually indistinguishable from the training environment described in the EMPO study (Zhang et al., 2025c), it provides a high-fidelity benchmark for assessing relative performance gains. This rigorous alignment of training conditions ensures that the superior results demonstrated by PowerFlow are attributable to the principled nature of our distribution matching framework rather than disparate optimization settings.

**Comparison with Temperature Scaling and Token-level Optimization.**    It is crucial to delineate the fundamental distinctions between PowerFlow and common strategies such as temperature scaling or token-level log-probability optimization. As established in recent analysis (Karan & Du, 2025), simply lowering the sampling temperature does not equate to sampling from a true power distribution $p^\alpha$ for $\alpha > 1$. Specifically, temperature scaling follows a conditional distribution $p_{temp}(x_t|x_{<t}) \propto (\sum_{x_{>t}} p(x_0, \ldots, x_T))^\alpha$, which averages future likelihoods in a greedy manner. In contrast, the power distribution targets $p_{pow}(x_t|x_{<t}) \propto \sum_{x_{>t}} p(x_0, \ldots, x_T)^\alpha$, thereby explicitly accounting for the sharpening of high-likelihood future paths—a property essential for surfacing correct reasoning trajectories. This theoretical gap explains why the *Low-temp* baseline in our experiments fails to match the robust elicitation achieved via principled distribution matching. Furthermore, treating the token-level average log-probability as a reward objective discards critical information regarding the global trajectory density. Such methods frequently succumb to local optima by over-exploiting low-entropy tokens, leading to the vacuous or repetitive generation observed in prior RLIF studies (Zhao et al., 2025; Ghimire et al., 2026) and illustrated in Figure 3. PowerFlow instead remains theoretically anchored in the global distribution $\pi^*(y|q) \propto p_{\text{base}}(y|q)^\alpha / Z'_\phi(q)^{|y|}$. By utilizing the $Z'_\phi(q)^{|y|}$ term to neutralize the structural length bias of autoregressive generation, our framework effectively filters the base model's density. Although this reparameterization does not strictly preserve mode rankings across varying sequence lengths, it largely maintains the model's semantic essence and relative structure while mitigating the degenerative biases inherent in generation probabilities. While this length-aware objective serves as a pragmatic compromise to neutralize the structural length bias of autoregressive models, it represents an initial step toward achieving true length invariance in distribution matching. We anticipate that future research will yield even more principled mechanisms for elegantly decoupling sequence length from semantic density.

**Relationship with the FlowRL Framework.**    While PowerFlow incorporates architectural insights from FlowRL, such as the use of an amortized partition function and importance sampling, there are fundamental distinctions in our theoretical motivation and treatment of sequence length. FlowRL employs GFlowNets within the paradigm of reinforcement learning from verifiable rewards (RLVR) to specifically mitigate the challenges of mode collapse and the over-optimization of dominant reward signals. In contrast, PowerFlow is formulated as a purely unsupervised framework for directional capability elicitation, aligning the policy with the intrinsic $\alpha$-power distribution of the base model itself. This conceptual shift fundamentally redefines the role of sequence length, transitioning it from a numerical stability concern into a structural alignment requirement. Specifically, in FlowRL, length normalization is primarily introduced as a reward-shaping technique to stabilize training and mitigate gradient explosion in long-trajectory reasoning. PowerFlow instead introduces a length-aware Trajectory-Balance (LA-TB) objective derived from a structural reparameterization of the energy surface, where the partition function is reformulated as $Z_\phi(q, y) = (Z'_\phi(q))^{|y|}$. This effectively projects the distribution matching problem onto a space of geometric mean probabilities. This reparameterization is not merely a numerical stabilizer but a theoretical necessity for robust, unsupervised distribution alignment; without it, the intensification of probability mass under $\alpha > 1$ would inevitably drive the model toward trivial, short sequences that exploit the exponential decay of path probabilities. Thus, PowerFlow establishes length invariance as a principled foundation for eliciting latent capabilities on a normalized energy landscape, moving beyond pragmatic modifications for optimization stability.

