# OpenReview forum: "PowerFlow: Unlocking the Dual Nature of LLMs via Principled Distribution Matching"
_ICML.cc/2026/Conference — ICML 2026 regular_

### Official Review · Reviewer_6RoD · 2026-03-12

**Soundness:** 3
**Presentation:** 3
**Significance:** 3
**Originality:** 3
**Overall Recommendation:** 4
**Confidence:** 3

**Summary:**

The paper introduces PowerFlow, a principled framework for unsupervised LLM fine-tuning that moves away from heuristic intrinsic rewards toward a distribution matching objective. By utilizing GFlowNets as amortized samplers, the authors target the $\alpha$-power (escort) distribution of the base model. This allows for directional capability elicitation through a single control knob, $\alpha$: sharpening ($\alpha > 1$) to concentrate mass on correct reasoning paths, or flattening ($\alpha < 1$) to recover creative diversity often suppressed by standard alignment.

A key technical contribution is the Length-Aware Trajectory-Balance (LA-TB) objective. This objective reparameterizes the partition function into an amortized token-level energy term, neutralizing the exponential length bias inherent in autoregressive generation that typically causes "length collapse" during sharpening or "repetitive explosion" during flattening.

**Compliance With Llm Reviewing Policy:**

Affirmed.

**Final Justification:**

The reviewer appreciates the rebuttal and has no further questions. I have decided to keep the original score.

**Key Questions For Authors:**

Beyond $\alpha$-Power: The paper mentions the framework is generalizable to other distribution families. What other distribution families could benefit from the training of LLMs for multi-step reasoning than a simple power transformation?

**Limitations:**

yes

**Strengths And Weaknesses:**

Strengths

Soundness & Originality: The framework is theoretically grounded, reformulating RLIF as a variational inference problem. By targeting $\alpha$-power distributions, it preserves the relative mode rankings and structural features of the base model, avoiding the biased distributional drift common in handcrafted rewards.

Technical Innovation: The LA-TB objective is an innnovative solution to the "structural length bias" of LLMs.

Significant Results: PowerFlow $(\alpha > 1)$ consistently outperforms strong baseline approaches in reasoning accuracy across multiple benchmarks (MATH, AIME, GPQA).

Pareto Improvements: On creative tasks, PowerFlow $(\alpha < 1)$ achieves a rare "best-of-both-worlds" result, shifting the Pareto frontier by improving both semantic diversity and output quality simultaneously.



Weaknesses
- Flexibility vs. Efficiency: While the approach is principled, the "control knob" $\alpha$ is fixed during training. This necessitates separate training runs for a "reasoning-heavy" model versus a "creative-heavy" model, whereas inference-time methods like temperature scaling are more flexible.

---

> ### Author Rebuttal · Authors · 2026-03-31
>
> Thank you for your time and effort in reviewing our paper! We are grateful for your constructive suggestions, which have significantly guided our improvements. Please find our responses to your comments below.
>
> ---
>
> **W1:** We appreciate this observation. Currently, different $\alpha$ values require separate training runs, whereas temperature can be adjusted at inference time. However, we note that the two operations target fundamentally different sequence-level distributions. As shown in [1, Proposition 1], low-temperature sampling does *not* sample from $p^\alpha$. The $\alpha$-power distribution computes next-token weights via a *sum of exponents* $p_{\text{pow}}(x_t|x_{<t}) \propto \sum_{x_{>t}} p(x_{0:T})^\alpha$, accounting for all future paths; temperature uses an *exponent of sums* $p_{\text{temp}}(x_t|x_{<t}) \propto \big(\sum_{x_{>t}} p(x_{0:T})\big)^\alpha$, averaging futures before sharpening. Thus $p^\alpha$ upweights tokens leading to higher-likelihood completions by planning ahead, while temperature amplifies locally popular tokens. This gap is confirmed empirically in both directions: our Low-temp baseline in Table 1 underperforms PowerFlow for reasoning ($\alpha>1$), and our High-temp baseline in Sec. 4.3 improves diversity only at the expense of quality for creativity ($\alpha<1$), while PowerFlow achieves a Pareto improvement. The flexibility of temperature therefore comes with a different—and in our setting, weaker—inductive bias.
>
> That said, we view developing inference-time $\alpha$ adjustment as a compelling direction. One path would be $\alpha$-conditioned training, potentially enabling a single model to support multiple generation modes. We leave this for future work.
>
> **Q1:** The TB objective can in principle be defined for any specified unnormalized target energy $\mathcal{E}(y, q)$ via $R(y) = \exp(-\mathcal{E}(y, q))$. The $\alpha$-power family was chosen for its theoretical elegance (the direct link to base model capabilities in Proposition 3.1) and its fully unsupervised nature. We did not evaluate the following variants in this paper, but mention them as natural extensions enabled by the framework:
>
> (1) **Reward-tilted distributions**: $\pi(y|q) \propto p_{\text{base}}(y|q)\,\exp(\beta\, r(y, q))$, where $r$ is an external signal such as a process reward model that scores intermediate reasoning steps. This bridges PowerFlow with reward-based methods while retaining the TB training objective. The $\alpha$-power case corresponds to $r = (\alpha-1)\log p_{\text{base}}$ with $\beta=1$.
>
> (2) **Step-wise transformations**: For multi-step tasks such as tool-augmented reasoning or agentic planning, one could apply different sharpening strengths at different stages—stronger concentration at decision points where future-path selection matters most, and milder transformation during execution. This would tailor the energy landscape to the hierarchical structure of sequential decision-making.
>
> (3) **Constraint-satisfying distributions**: $\pi(y|q) \propto p_{\text{base}}(y|q)^\alpha \cdot \prod_k c_k(y)$, where $c_k$ are soft constraint functions (e.g., format compliance, tool-call validity, safety). The TB objective handles unnormalized targets natively, so the intractable partition function poses no additional difficulty. This product-of-experts formulation could enable multi-objective generation within a single model.
>
> We believe exploring these families is a promising direction enabled by our framework's generality.
>
> ---
>
> We hope these clarifications address your concerns. If so, we wonder if you could kindly consider raising your score? We will also be happy to answer any further questions you may have. Thank you very much!
>
> [1] Reasoning with Sampling: Your Base Model is Smarter Than You Think, ICLR 2026 Oral.

---

> > ### Author Rebuttal · Reviewer_6RoD · 2026-04-03
> >
> > The reviewer appreciates the rebuttal and has no further questions. I have decided to keep the original score.

---

> > > ### Author Response · Authors · 2026-04-04
> > >
> > > Thank you for your thoughtful review and for maintaining a positive score! We are glad that our responses have fully addressed your concerns. We would like to briefly share several additional improvements we made during the rebuttal period that were not included in our response to you:
> > >
> > > ---
> > >
> > > 1. **New theoretical results.** We proved that LA-TB is the *I-projection* of $p_\alpha$ onto length-calibrated distributions, i.e., the unique solution to $\min\_{\pi} D\_{\mathrm{KL}}(\pi(\cdot|q) \| p\_\alpha(\cdot|q)) \;\text{s.t.}\; \mathbb{E}\_\pi[L] = \ell\_q$ is $\pi^\star(y|q) \propto p\_\alpha(y|q) e^{-\lambda\_q L(y)}$, where the learned $\log Z'(q)$ plays exactly the role of the Lagrange multiplier $\lambda_q$. This means Eq. (8) is the *minimum-distortion* correction to $p_\alpha$ among all distributions with calibrated mean length. We further established that: (i) conditional on length, LA-TB preserves the $\alpha$-power distribution *exactly*, i.e., $\pi_\lambda(y|q, L\!=\!m) = p_\alpha(y|q, L\!=\!m)$ for all $m$; (ii) for any pair $(y_i, y_j)$, the log-ratio distortion follows $\log\frac{\pi_\lambda(y_i|q)}{\pi_\lambda(y_j|q)} = \log\frac{p_\alpha(y_i|q)}{p_\alpha(y_j|q)} - \lambda_q(L(y_i) - L(y_j))$, so a rank inversion can only occur when the original $\alpha$-power log-margin is smaller than $|\lambda_q| \cdot |\Delta L|$. On Qwen2.5-Math-1.5B, the empirical pairwise inversion rate is only \~9% (\~91% of rankings preserved).
> > >
> > > 2. **Comprehensive ablation study.** On Qwen2.5-Math-1.5B:
> > >
> > > | Variant | Avg | SEM$_{\text{avg}}$ |
> > > |---|---|---|
> > > | PowerFlow w/o length correction & format penalty (= TB-traj, Fig. 3) | ≈ 0 | — |
> > > | PowerFlow w/o length correction | 14.30 | 0.27 |
> > > | PowerFlow w/o format penalty | 33.48 | 0.36 |
> > > | **PowerFlow (full)** | **34.30** | **0.30** |
> > >
> > > This cleanly isolates the contribution of each component.
> > >
> > > 3. **Statistical significance.** We computed SEM$_{\text{avg}}$ (standard error of the mean, averaged across 6 benchmarks, ×100 for percentage scale):
> > >
> > > | Model | Method | Avg | SEM$_{\text{avg}}$ |
> > > |---|---|---|---|
> > > | Qwen2.5-1.5B | PowerFlow | 19.85 | 0.28 |
> > > | Qwen2.5-1.5B | GRPO | 18.13 | 0.29 |
> > > | Qwen2.5-Math-1.5B | PowerFlow | 34.30 | 0.30 |
> > > | Qwen2.5-Math-1.5B | GRPO | 32.75 | 0.31 |
> > > | Qwen2.5-Math-7B | PowerFlow | 42.17 | 0.27 |
> > > | Qwen2.5-Math-7B | GRPO | 42.38 | 0.38 |
> > > | Llama-3.2-3B | PowerFlow | 22.88 | 0.33 |
> > > | Llama-3.2-3B | GRPO | 22.33 | 0.33 |
> > >
> > > Performance differences exceed SEM$_{\text{avg}}$ in all cases where we claim improvement.
> > >
> > > ---
> > >
> > > We believe these additions have meaningfully strengthened the paper's rigor and completeness. We would be very grateful if you could kindly consider whether a score increase might be warranted. We fully respect your assessment. Thank you again for your detailed and constructive feedback!

---

### Official Review · Reviewer_hNo9 · 2026-03-12

**Soundness:** 3
**Presentation:** 3
**Significance:** 3
**Originality:** 3
**Overall Recommendation:** 4
**Confidence:** 3

**Summary:**

This paper introduces PowerFlow, an unsupervised fine-tuning framework that frames the alignment of Large Language Models as a distribution matching problem. Moving away from heuristic intrinsic rewards commonly used in RLIF, PowerFlow targets the alpha-power distribution of the base model. To combat the structural length bias and optimization instability inherent in autoregressive generation under standard Trajectory Balance, the authors propose a Length-Aware Trajectory-Balance (LA-TB) objective. By treating the power exponent alpha as a control parameter, the framework aims to directionally elicit capabilities: sharpening the distribution (alpha > 1) to enhance logical reasoning, or flattening it (alpha < 1) to unlock creative diversity in aligned models. The authors evaluate the method across multiple model scales, demonstrating improvements over existing RLIF baselines in mathematical reasoning and Pareto improvements in creative writing tasks.

**Compliance With Llm Reviewing Policy:**

Affirmed.

**Key Questions For Authors:**

1. Theoretical Consistency: You motivate the alpha-power target by its ability to strictly preserve relative mode rankings. However, your LA-TB objective converges to a length-normalized distribution (Equation 8), which you admit does not strictly maintain these rankings. Can you mathematically formalize or empirically quantify how much this length penalization warps the actual mode rankings? How does this not degrade your method into another heuristically biased approach?
2. Baseline Fairness: I acknowledge your note in Appendix F regarding computational constraints dictating the use of open-source RLIF checkpoints. However, how can you guarantee that the performance gap between PowerFlow and models like EMPO or TTRL is not simply an artifact of differences in training data mixtures (e.g., exact subset of NuminaMath-CoT) or prompt formatting applied during the creation of those open-source checkpoints?
3. GRPO Comparison: Table 1 shows that on Qwen2.5-Math-7B, PowerFlow and GRPO perform comparably, with GRPO still leading on AIME24. Given PowerFlow's advantage is most pronounced on smaller models (1.5B/3B), isn't it likely that this reflects GRPO's optimization instability at smaller scales rather than PowerFlow fundamentally surpassing external verifiable rewards? Figure 7's pass@n analysis is helpful, but it only compares against the Base model. Could you provide a direct pass@n comparison between GRPO and PowerFlow on the 1.5B/3B scale? If the gap narrows at high n, it would support the optimization instability hypothesis; if it persists, how would you reinterpret PowerFlow's advantage?
4. Initialization Heuristics: The log Z' module relies on a specialized initialization using the reference model's average token-level log-probability to prevent gradient fluctuations. How sensitive is the framework's overall convergence and final performance to this specific initialization strategy? Does it fail without it?

**Limitations:**

yes

**Strengths And Weaknesses:**

Soundness：
- Strengths: The paper successfully identifies and demonstrates the vulnerability of standard RL and GFlowNet objectives to length collapse when applied to autoregressive LLMs without external rewards. The empirical evaluation spans multiple model scales (1.5B to 32B) and distinct domains (mathematical reasoning and creative writing), providing a comprehensive view of the framework's versatility.

- Weaknesses: There is a noticeable theoretical tension between the paper's core motivation and its practical implementation. The paper heavily motivates the use of the alpha-power distribution on the premise that it strictly preserves relative probability rankings and mode structure, thereby avoiding heuristic biases. However, to mitigate structural length bias, the proposed LA-TB objective (Equation 7) introduces an amortized token-level energy term, forcing convergence to a length-normalized target distribution (Equation 8). While the authors candidly acknowledge in Section 3.3 and Appendix F that this "does not strictly maintain relative mode rankings," this structural reparameterization still functions as a de facto heuristic length penalty. This fundamental compromise directly challenges the paper's central claim of achieving a purely "principled" and unbiased distribution matching, revealing an inherent tension between theoretical motivation and practical necessity.
- Furthermore, the baseline comparisons present potential fairness issues despite the authors' clarifications. I note the discussion in Appendix F, where the authors explain that computational constraints necessitated the use of official open-source checkpoints for RLIF baselines (Intuitor, EMPO, TTRL). While they made efforts to train their internal GRPO baseline using a configuration identical to PowerFlow, comparing internally trained models against external checkpoints trained with potentially different data mixtures, formatting templates, and hyperparameters inevitably introduces confounding variables. This makes the empirical claims of clear superiority over SOTA RLIF methods somewhat precarious.
- Finally, the claim that unsupervised PowerFlow broadly surpasses supervised GRPO requires more nuanced contextualization. Table 1 shows that on the most capable model tested (Qwen2.5-Math-7B), the performance between PowerFlow and GRPO is highly competitive but mixed (averaging 42.17 vs 42.38). While PowerFlow holds an edge on AIME25 and GPQA, GRPO maintains superiority on critical benchmarks like AIME24 (22.70 vs 20.00) and MATH500. Given this parity at the 7B scale, it is highly probable that PowerFlow's more pronounced outperformance on smaller models (1.5B/3B) is partially attributable to the severe RL optimization instability (alignment tax) typical of GRPO at those constrained scales, rather than PowerFlow fundamentally eclipsing verifiable external rewards.

Presentation:
- Strengths: The narrative is compelling, and the manuscript is generally well-structured. The conceptual visualization of the "dual nature" (Figure 1) and the optimization stability analysis (Figure 3) effectively communicate the motivation and the mechanics of the proposed solution.
- Weaknesses: Crucial details regarding the GRPO baseline are omitted from the main text and relegated to the appendices. Given that comparing an unsupervised method to a supervised baseline is a central claim of the paper, the exact nature of the verifiable rewards used for GRPO must be explicitly detailed in the main experimental setup to ensure transparency.

Significance:
- Strengths: Proposing a unified mechanism (the alpha knob) to modulate both reasoning extraction and creative diversity without relying on external reward models is a highly relevant and impactful concept for the alignment community.
- Weaknesses: The significance of the contribution is dampened by the theoretical compromises made in the LA-TB objective and the potential fairness issues in the empirical comparisons with RLIF baselines and the mixed results against the supervised GRPO baseline.

Originality:
- Strengths: While previous works have explored GFlowNets for LLMs (e.g., FlowRL), the explicit use of the escort distribution (alpha-power distribution) to bidirectionally control entropy for unsupervised capability elicitation is a creative and novel synthesis of statistical mechanics and LLM alignment.

---

> ### Author Rebuttal · Authors · 2026-03-31
>
> Thank you for your time and effort in reviewing our paper! We are grateful for your constructive suggestions, which have significantly guided our improvements. Please find our responses to your comments below.
>
> ---
>
> **W1/Q1:** We appreciate this important question. We provide a new theorem showing that LA-TB is the *I-projection* of $p_\alpha$ onto length-calibrated distributions:
>
> $$\min_{\pi} D_{\mathrm{KL}}(\pi(\cdot|q) \| p_\alpha(\cdot|q)) \quad \text{s.t.} \quad \mathbb{E}_\pi[L] = \ell_q,$$
>
> whose unique solution is $\pi^\star(y|q) \propto p_\alpha(y|q) e^{-\lambda_q L(y)}$, where $\lambda_q$ is the Lagrange multiplier chosen so that $\mathbb{E}\_{\pi^\star}[L]=\ell\_q$. In our method, the learned $\log Z'(q)$ plays exactly the role of $\lambda_q$. This means Eq. (8) is the *minimum-distortion* correction to $p_\alpha$ among all distributions with calibrated mean length.
>
> To quantify the impact on mode rankings, we further provide two propositions. First, conditional on length, LA-TB preserves the $\alpha$-power distribution *exactly*:
>
> $$\pi_\lambda(y|q, L\!=\!m) = p_\alpha(y|q, L\!=\!m), \quad \forall\, m.$$
>
> Second, for any pair $(y_i, y_j)$, the log-ratio distortion is:
>
> $$\log\frac{\pi_\lambda(y_i|q)}{\pi_\lambda(y_j|q)} = \log\frac{p_\alpha(y_i|q)}{p_\alpha(y_j|q)} - \lambda_q(L(y_i) - L(y_j)).$$
>
> Thus, a rank inversion can occur *only* when the $\alpha$-power log-margin $< |\lambda_q| \cdot |\Delta L|$. On Qwen2.5-Math-1.5B, the pairwise inversion rate $\mathrm{IR} \approx 0.09$ (~91% of rankings preserved). The distortion is therefore confined to cross-length comparisons and is empirically small. We will add these propositions and proofs in the revision.
>
> **W2/Q2:** We appreciate this concern. Beyond our internally trained GRPO baseline (which shares the exact same data, prompts as PowerFlow), our training setup also aligns with EMPO [1] on training data (the same 18K NuminaMath-CoT subset), prompts, Clip-Higher,  hyperparameters, and format requirements, enabling a controlled comparison with EMPO's open-source checkpoint. PowerFlow consistently outperforms EMPO across all four model variants. For Intuitor and TTRL, reproducing their full training pipelines exceeds our compute budget. We compare against their official checkpoints and view these comparisons as indicative. We will clarify setup differences more clearly in the revision.
>
> **W3/Q3:** We provide the requested pass@n comparison between PowerFlow and GRPO on Qwen2.5-1.5B (OlympiadBench, anonymous link: https://imgur.com/a/Fn7tACn). The two methods converge at high $n$, suggesting they access similar latent solution spaces. The key difference lies in sample efficiency: PowerFlow achieves substantially better pass@1 through principled density matching over the full energy landscape. On smaller models, GRPO relies on sparse binary rewards (correct/incorrect), which provide a noisy optimization signal with high gradient variance at limited parameter scales. By reshaping the $\alpha$-power energy surface smoothly, PowerFlow provides a denser and more stable training signal, and this stable optimization is *exactly* an advantage of our method. At 7B where model capacity is sufficient, the two methods perform comparably. We emphasize that PowerFlow achieves competitive or superior performance entirely *without* external rewards.
>
> **W4:** We apologize for this presentation oversight. For GRPO, we follow the DAPO [2] paradigm and verl's recipe, using answer-verification rewards that check whether the final answer in \boxed{} matches the ground truth. We also adopt Clip-Higher (asymmetric clipping to encourage exploration), which is shared with PowerFlow training. We will move these details from the appendix to the main experimental setup for transparency.
>
> **W5:** The theoretical concern is addressed in W1/Q1: LA-TB is the minimum-distortion I-projection, with within-shell rankings exactly preserved and cross-length distortion empirically small. The baseline fairness concern is addressed in W2/Q2 via EMPO alignment. Together with the new pass@n evidence in W3/Q3, we believe the contribution remains significant.
>
> **Q4:** Without the specialized initialization, the initial variations of $\log Z'$ are larger, extending the warm-up phase by ~10 gradient steps. However, training stabilizes normally afterward and final performance is unaffected in our experiments across all tested models. The initialization is a practical convenience that accelerates early convergence, not a requirement for the method to work.
>
> ---
>
> We hope these clarifications address your concerns. If so, we wonder if you could kindly consider raising your score? We will also be happy to answer any further questions you may have. Thank you very much!
>
> [1] Right Question is Already Half the Answer: Fully Unsupervised LLM Reasoning Incentivization, NeurIPS 2025 Spotlight.
>
> [2] DAPO: An Open-Source LLM Reinforcement Learning System at Scale, NeurIPS 2025.

---

> > ### Author Rebuttal · Reviewer_hNo9 · 2026-04-03
> >
> > The author has resolved my issue and I will keep my positive score.

---

> > > ### Author Response · Authors · 2026-04-04
> > >
> > > Thank you for your rigorous and insightful review and for maintaining a positive score! We are glad that our responses have fully addressed your concerns. We would like to briefly highlight two additional results from the rebuttal period that were not included in our direct response to you:
> > >
> > > ---
> > >
> > > 1. **Comprehensive ablation study.** On Qwen2.5-Math-1.5B:
> > >
> > > | Variant | Avg | SEM$_{\text{avg}}$ |
> > > |---|---|---|
> > > | PowerFlow w/o length correction & format penalty (= TB-traj, Fig. 3) | ≈ 0 | — |
> > > | PowerFlow w/o length correction | 14.30 | 0.27 |
> > > | PowerFlow w/o format penalty | 33.48 | 0.36 |
> > > | **PowerFlow (full)** | **34.30** | **0.30** |
> > >
> > > This cleanly isolates the contribution of each design choice and reinforces that our gains are not attributable to any single component.
> > >
> > > 2. **Statistical significance.** We computed SEM$_{\text{avg}}$ (standard error of the mean, averaged across 6 benchmarks, ×100 for percentage scale):
> > >
> > > | Model | Method | Avg | SEM$_{\text{avg}}$ |
> > > |---|---|---|---|
> > > | Qwen2.5-1.5B | PowerFlow | 19.85 | 0.28 |
> > > | Qwen2.5-1.5B | GRPO | 18.13 | 0.29 |
> > > | Qwen2.5-Math-1.5B | PowerFlow | 34.30 | 0.30 |
> > > | Qwen2.5-Math-1.5B | GRPO | 32.75 | 0.31 |
> > > | Qwen2.5-Math-7B | PowerFlow | 42.17 | 0.27 |
> > > | Qwen2.5-Math-7B | GRPO | 42.38 | 0.38 |
> > > | Llama-3.2-3B | PowerFlow | 22.88 | 0.33 |
> > > | Llama-3.2-3B | GRPO | 22.33 | 0.33 |
> > >
> > > Performance differences exceed SEM$_{\text{avg}}$ in all cases where we claim improvement, confirming statistical significance.
> > >
> > > ---
> > >
> > > We believe these supplementary results have meaningfully strengthened the paper. We would be very grateful if you could kindly consider whether a score increase might be warranted. We fully respect your assessment. Thank you again for your detailed and constructive feedback!

---

### Official Review · Reviewer_q98o · 2026-03-12

**Soundness:** 2
**Presentation:** 2
**Significance:** 2
**Originality:** 3
**Overall Recommendation:** 4
**Confidence:** 3

**Summary:**

The paper explores the topic of unsupervised elicitation of the latent models capabilities with Reinforcement Learning from Internal Feedback (RLIF). Authors propose a new approach for RLIF PowerFlow. PowerFlow views the RLIF objective through the lens of distribution matching and tries to align model distribution with \alpha-powered distribution. \alpha-powered distribution gives a natural way of balancing between generation diversity and distribution sharpening. Additionally, authors extend distribution matching loss with terms to account for length bias and formal correction of generations.

**Compliance With Llm Reviewing Policy:**

Affirmed.

**Final Justification:**

Authors resolved the concerns raised during the review phase clearly, therefore I increase my score from 3 to 4.

**Key Questions For Authors:**

1. Described motivation behind \alpha-power distribution and the role of \alpha in it is really similar to temperature. What are the differences?
2. Proposition 3.1 has a lot of missing context
3. Do you optimize objective (9) w.r.t. \phi and \varphi with GD methods? If yes, it is a parabola wrt to log Z, so you can calculate its minimum explicitly, have you tried it?
4. On line 259 you say ‘does not strictly maintain relative mode rankings’. How drastically mode rankings are affected, since maintaining it was one of the motivations behind \alpha-power distribution?
5. Among the baselines you mention format-only, however it appears only for one model. Why is that?
6. Ablation (or other baselines) is lacking. I think it is important to show the effect of different parts of your loss: only distribution matching, distribution matching + length correction, distribution matching + format-only.

**Limitations:**

yes

**Strengths And Weaknesses:**

Strengths:
- 1 Idea. Novel approach for RLIF. Moreover, authors combined it with different tricks to account for different issues during training.
- 2 Diversity analysis. Authors recognize potential problems with RL and diversity collapse and analyse it separately.

Weaknesses:
- 1 Clarity. The paper sometimes lacks clarity. See questions for all examples, but to name some: experimental setup for 4.3 is never described; proposition 3.1 operates with new entities but never introduces them; some important information hidden in appendix, for example the fact that some of baselines are open-source checkpoints that could’ve been trained in different settings
- 2 Evaluation results. On top of potential issues with different training setups for baselines, the main metric is an average of 6 datasets with different scales. Moreover, results seem super close, so confidence intervals or anything else to show significance or improvements is missing.
- 3 Strong wording for claims. Authors claim that existing approaches are based on some heuristics and in contrast PowerFlow to them. While initial motivation by distribution matching can be considered (arguable) as a natural objective, with all modifications to the loss it also becomes another heuristic.

---

> ### Author Rebuttal · Authors · 2026-03-31
>
> Thank you for your time and effort in reviewing our paper! We are grateful for your constructive suggestions, which have significantly guided our improvements. Please find our responses to your comments below.
>
> ---
>
> **W1/Q2:** The experimental details for Sec. 4.3 are in Sec. 4.1. For Proposition 3.1, we acknowledge omitting the definitions of $P_F$ (forward policy) and $P_B$ (backward policy). As stated in the main text (lines 306–309), we use open-source RLIF checkpoints. Our setup aligns with EMPO [1] in training data, prompts, hyperparameters and format for fair comparison. We will add this context to the main text.
>
> **W2:** We report the standard error of the mean (SEM$_{\text{avg}}$; 16 samples/problem, averaged across 6 benchmarks; $\times 100$ for percentage scale):
>
> | Model | Method | Avg | SEM$_{\text{avg}}$ |
> |---|---|---|---|
> | Qwen2.5-1.5B | PowerFlow | 19.85 | 0.28 |
> | Qwen2.5-1.5B | GRPO | 18.13 | 0.29 |
> | Qwen2.5-Math-1.5B | PowerFlow | 34.30 | 0.30 |
> | Qwen2.5-Math-1.5B | GRPO | 32.75 | 0.31 |
> | Qwen2.5-Math-7B | PowerFlow | 42.17 | 0.27 |
> | Qwen2.5-Math-7B | GRPO | 42.38 | 0.38 |
> | Llama-3.2-3B | PowerFlow | 22.88 | 0.33 |
> | Llama-3.2-3B | GRPO | 22.33 | 0.33 |
>
> Where we claim improvement, the gap exceeds SEM$_{\text{avg}}$. For Qwen2.5-Math-7B (comparable claim), the gap (0.21) is well within SEM. We will include these in the revision.
>
> **W3:** We provide a new theorem to clarify that the length-aware objective is not ad-hoc: it is the *I-projection* of $p_\alpha$ onto length-calibrated distributions:
>
> $$\min_{\pi} D_{\mathrm{KL}}(\pi(\cdot|q) \| p_\alpha(\cdot|q)) \quad \text{s.t.} \quad \mathbb{E}_\pi[L] = \ell_q,$$
>
> whose unique solution is $\pi^\star(y|q) \propto p_\alpha(y|q) e^{-\lambda_q L(y)}$, where $\lambda_q$ is the Lagrange multiplier chosen so that $\mathbb{E}\_{\pi^\star}[L]=\ell\_q$. In our method, the learned $\log Z'(q)$ plays exactly the role of $\lambda_q$. Thus Eq. (8) is the *minimum-distortion* correction to $p_\alpha$ under a mean-length constraint. We will add this in the revision.
>
> **Q1:** As shown in [2, Proposition 1], low-temperature sampling does *not* sample from $p^\alpha$. The power distribution computes next-token weights via a *sum of exponents* $p_{\text{pow}}(x_t|x_{<t}) \propto \sum_{x_{>t}} p(x_{0:T})^\alpha$, accounting for all future paths; temperature uses an *exponent of sums* $p_{\text{temp}}(x_t|x_{<t}) \propto \big(\sum_{x_{>t}} p(x_{0:T})\big)^\alpha$, greedily averaging futures. Thus, $p^\alpha$ upweights tokens with fewer but higher-likelihood futures, while temperature amplifies locally popular tokens. Table 1's Low-temp baseline confirms this.
>
> **Q3:** Yes, $\theta$ and $\phi$ are jointly optimized via GD. $Z'_\phi(q)$ is a neural network (3-layer MLP; Appendix B.1), not a per-prompt scalar. While the per-sample closed-form minimizer is trivially computable, the true $Z'^*(q)$ involves an intractable sum over all responses. The network learns to approximate it via amortization — the per-sample closed form would overfit individual samples.
>
> **Q4:** We formalize the ranking distortion via new propositions. Let $\lambda_q = \log Z'(q)$. Conditional on length, LA-TB preserves the $\alpha$-power distribution *exactly*:
>
> $$\pi_\lambda(y|q, L\!=\!m) = p_\alpha(y|q, L\!=\!m), \quad \forall\, m.$$
>
> Second, for any pair $(y_i, y_j)$, the log-ratio distortion is:
>
> $$\log\frac{\pi_\lambda(y_i|q)}{\pi_\lambda(y_j|q)} = \log\frac{p_\alpha(y_i|q)}{p_\alpha(y_j|q)} - \lambda_q(L(y_i) - L(y_j)).$$
>
> Thus, a rank inversion occurs *only* when the $\alpha$-power log-margin $< |\lambda_q| \cdot |\Delta L|$. On Qwen2.5-Math-1.5B, the pairwise inversion rate $\mathrm{IR} \approx 0.09$ (~91% preserved). We will include these in the revision.
>
> **Q5:** We now add a format-only experiment on Qwen2.5-Math-7B: avg performance 36.22 (SEM$_{\text{avg}}$=0.30), which is 5.95 points below PowerFlow (42.17). Combined with the 1.5B result (29.92 vs. 34.30), this confirms format compliance alone cannot explain our gains.
>
> **Q6:** We provide the requested ablation on Qwen2.5-Math-1.5B:
>
> | Variant | Avg | SEM$_{\text{avg}}$ |
> |---|---|---|
> | Dist. matching only (=TB-traj, Fig. 3) | $\approx 0$ | — |
> | Dist. matching + format penalty | 14.30 | 0.27 |
> | Dist. matching + length correction | 33.48 | 0.36 |
> | **PowerFlow (full)** | **34.30** | **0.30** |
>
> Without length correction, pure $\alpha$-power matching suffers catastrophic length collapse ($\to 0$, Fig. 3). Length correction is the critical component; format penalty provides additional gain via answer extraction.
>
> ---
>
> We hope our response addresses your concerns. If so, we wonder if you could kindly consider raising your score? We will also be happy to answer any further questions you may have. Thank you very much!
>
> [1] Right Question is Already Half the Answer: Fully Unsupervised LLM Reasoning Incentivization, NeurIPS 2025 Spotlight.
>
> [2] Reasoning with Sampling: Your Base Model is Smarter Than You Think, ICLR 2026 Oral.

---

> > ### Author Rebuttal · Reviewer_q98o · 2026-04-04
> >
> > Thank you for the detailed response, which resolves all major concerns about the work. Therefore, I'll increase my score from 3 to 4.

---

> > > ### Author Response · Authors · 2026-04-04
> > >
> > > Thank you for your careful review and for raising your score to 4! We are glad that our responses have fully addressed your concerns, and we are grateful for the constructive feedback that pushed us to improve the work. Thank you again for your valuable feedback!

---

### Official Review · Reviewer_YKpq · 2026-03-13

**Soundness:** 3
**Presentation:** 4
**Significance:** 3
**Originality:** 3
**Overall Recommendation:** 4
**Confidence:** 3

**Summary:**

This paper proposes PowerFlow, a framework for unsupervised fine-tuning of large language models by framing post-training as a distribution matching problem. Instead of designing heuristic intrinsic rewards as in prior Reinforcement Learning from Internal Feedback (RLIF) methods, the method targets an alpha-power transformation of the base model distribution. The approach interprets GFlowNets as amortized samplers of an unnormalized density and derives a length-aware trajectory-balance objective designed to mitigate structural length bias in autoregressive generation. The framework allows directional control via the exponent alpha: sharpening the distribution (alpha > 1) to emphasize reasoning trajectories and flattening it (alpha < 1) to increase diversity and creativity. Experiments across reasoning benchmarks and creative writing tasks suggest the method can outperform prior RLIF approaches and achieve results competitive with supervised RL methods such as GRPO while preserving diversity.

The authors strive to address the concept of unsupervised capability elicitation by grounding RLIF-style training in a principled probabilistic objective rather than heuristic rewards. Overall, the article's specific domain pertains to unsupervised alignment and post-training optimization for large language models.

**Compliance With Llm Reviewing Policy:**

Affirmed.

**Key Questions For Authors:**

Sensitivity to the alpha parameter
The method relies on the alpha exponent as a key control parameter to elicit reasoning (alpha > 1) or creativity (alpha < 1). How sensitive is performance to the choice of alpha across models, tasks, and training regimes? In particular, do the authors observe large variance if alpha is slightly mis-specified, and is there a principled way to select or schedule alpha during training? Clarifying this would affect my evaluation of the method’s robustness and practical usability.

Applicability beyond reasoning and creative generation
Most experiments focus on mathematical reasoning and creative writing tasks. Have the authors evaluated the method on other categories such as factual QA, tool-use tasks, agentic planning, or coding benchmarks? Demonstrating effectiveness across a broader set of capabilities would strengthen the claim that distribution matching provides a general mechanism for capability elicitation.

Scalability and compute overhead
The approach introduces a GFlowNet-style objective with an amortized partition function. How does the training cost compare with standard RLIF methods or supervised RL methods such as GRPO in practice (e.g., number of samples, stability, wall-clock time)? Understanding the computational trade-offs would influence my assessment of the method’s practicality for large-scale models.

Effect when the base model lacks latent capability
The framework preserves the relative structure of the base model distribution through alpha-power transformations. How does PowerFlow behave when the base model does not already contain strong latent reasoning capability? For example, does sharpening simply amplify incorrect modes, or can the method still produce meaningful improvements? Evidence or discussion here would clarify the limits of the approach.

Stability of the length-aware objective
The paper argues that the length-aware trajectory balance objective resolves structural length bias. Could the authors provide additional empirical evidence (e.g., ablations across different sequence length distributions or tasks) demonstrating that the objective consistently prevents length collapse or repetitive generation across model scales? This would help confirm that the proposed solution generalizes beyond the specific experimental setup.

**Limitations:**

yes

**Strengths And Weaknesses:**

Strengths

Clear conceptual framing. The paper presents a coherent reinterpretation of unsupervised post-training as a distribution matching objective. The alpha-power formulation provides a clean theoretical lens connecting sharpening, diversity control, and RL-style training.

Principled treatment of length bias. The proposed length-aware trajectory balance objective addresses a known pathology of autoregressive likelihood objectives, namely their implicit preference for shorter sequences. The analysis linking trajectory probabilities to length collapse is well motivated.

Unified control mechanism. The idea of using the alpha exponent as a single knob to control reasoning vs. creativity is conceptually appealing and offers a simple mechanism for capability modulation.

Strong empirical comparisons. Experiments evaluate several model families and compare against multiple RLIF baselines as well as supervised RL methods. Results indicate competitive or improved performance in several settings.

Diversity preservation analysis. The paper evaluates reasoning diversity and shows that the method maintains multiple solution strategies rather than collapsing to a single trajectory, which is a known issue with RL training.

Weaknesses

Limited theoretical guarantees. While the distribution matching framework is well motivated, the paper does not fully characterize when alpha-power matching improves reasoning accuracy. The theoretical arguments largely remain intuitive rather than providing formal guarantees about capability elicitation.

Dependence on the base distribution. Because the method preserves the relative structure of the base model distribution, improvements rely on latent capabilities already present in the pretrained model. The approach may therefore have limited impact when the base model lacks relevant knowledge.

Evaluation scope. Most experiments focus on mathematical reasoning and creative writing. It is unclear whether the same mechanism generalizes to broader reasoning tasks such as tool use, multi-step planning, or agentic tasks.

Sensitivity to alpha. Although the method introduces alpha as a key control parameter, the paper does not systematically explore sensitivity or provide guidance for selecting alpha across tasks or model scales.

Training complexity and scalability. The use of GFlowNet-style objectives and amortized partition function estimation may introduce additional optimization complexity. The paper does not fully discuss training stability, compute cost, or scalability to larger frontier models.

---

> ### Author Rebuttal · Authors · 2026-03-31
>
> Thank you for your time and effort in reviewing our paper! We are grateful for your constructive suggestions, which have significantly guided our improvements. Please find our responses to your comments below.
>
> ---
>
> **W1:** We indeed follow the assumption—supported by a growing body of evidence—that reasoning can be elicited through distribution sharpening [1–3], and our empirical results validate this across 4 model families. We acknowledge that formally characterizing *when* such sharpening is beneficial is an important and interesting future direction. Notably, [2] formalizes conditions under which sharpening succeeds via a *coverage coefficient*, and [1] shows RL-trained models remain bounded by the base model's latent capability. These results suggest that key preconditions include non-negligible base model mass on correct solutions and a verification-generation gap. We will add a formal discussion of these conditions in the revision.
>
> **W2/Q4:** From an information-theoretic view, sharpening primarily amplifies existing probability mass—it does not create new knowledge if the base model assigns negligible mass to correct solutions. Our pass@$n$ analysis (Appendix D.2) confirms this: both PowerFlow and GRPO converge to base model performance as $n\to 256$, suggesting improved sampling efficiency rather than novel capabilities. When the base model genuinely lacks relevant knowledge, sharpening will amplify incorrect modes—a limitation shared by current distribution-sharpening approaches [2, 4]. Nonetheless, when the base model has non-negligible coverage over correct solutions, PowerFlow provides a principled and fully unsupervised way to elicit these capabilities, as our results across 4 model families confirm. We will add an explicit discussion of these boundary conditions in the revision.
>
> **W3/Q2:** We agree that extending to tool use, planning, and agentic tasks is a meaningful future direction. Beyond math and creative writing, our evaluation includes GPQA (graduate-level science reasoning), covering both modes of $\alpha$-power: sharpening ($\alpha>1$) and flattening ($\alpha<1$). We believe the framework will benefit other tasks requiring logical structure or creativity, and leave broader evaluation for future work.
>
> **W4/Q1:** Appendix D.1 reports sensitivity on Qwen2.5-Math-1.5B: $\alpha\in\{2,4,6\}$ yields avg performance 34.08, 34.30, 33.64—fluctuation $<0.7$ across a 3× range on this model. For instruct models, we empirically find $\alpha=2$ more effective, likely because alignment has already concentrated the distribution (Sec. 4.2). While performance is robust within the tested range, we did not perform exhaustive per-model $\alpha$ tuning—the reported results use a single default per model type (base vs. instruct), which we view as evidence of practical robustness. As stated in Sec. 4.2, we leave the development of automated $\alpha$-adjustment mechanisms for future work.
>
> **W5/Q3:** The Log-Partition Function Estimator is a 3-layer MLP (Appendix B.1) adding negligible parameters to the LLM. Per-step overhead comes from computing base model log-probabilities on sampled trajectories: in our 7B setup on 8×H100, wall-clock time increases from 100.1s (GRPO) to 109.7s (PowerFlow) per step. In our recommended setting with a 3× larger initial LR, PowerFlow converges in comparable steps to GRPO with stable training curves (Fig. 3). We will include these details in the revision.
>
> **Q5:** We provide response length curves for all four models during training at https://imgur.com/a/E4BYHgq, showing that lengths remain stable throughout optimization without explosion or collapse across model scales (1.5B–7B). Theoretically, this is consistent with a new theorem we provide: LA-TB is the *I-projection* of $p_\alpha$ onto length-calibrated distributions:
>
> $$\min_{\pi} D_{\mathrm{KL}}(\pi(\cdot|q) \| p_\alpha(\cdot|q)) \quad \text{s.t.} \quad \mathbb{E}_\pi[L] = \ell_q,$$
>
> whose unique solution is $\pi^\star(y|q) \propto p_\alpha(y|q) e^{-\lambda_q L(y)}$, where $\lambda_q$ is the Lagrange multiplier chosen so that $\mathbb{E}\_{\pi^\star}[L]=\ell\_q$. In our method, the learned $\log Z'(q)$ plays exactly the role of $\lambda_q$. This means Eq. (8) is the *minimum-distortion* correction to $p_\alpha$ under a mean-length constraint, rather than an arbitrary heuristic bias. We will include this theorem in the revision.
>
> ---
>
> We hope these clarifications address your concerns. If so, we wonder if you could kindly consider raising your score? We will also be happy to answer any further questions you may have. Thank you very much!
>
> [1] Does Reinforcement Learning Really Incentivize Reasoning Capacity in LLMs Beyond the Base Model?, NeurIPS 2025 Oral.
>
> [2] Self-Improvement in Language Models: The Sharpening Mechanism, ICLR 2025 Oral.
>
> [3] Reasoning with Sampling: Your Base Model is Smarter Than You Think, ICLR 2026 Oral.
>
> [4] How Far Can Unsupervised RLVR Scale LLM Training?, ICLR 2026.

---

### Decision · Program_Chairs · 2026-04-30

**Decision:**

Accept (regular)

**Comment:**

Key Reasons for Acceptance:
* Theoretical Rigor: The proposed Length-Aware Trajectory-Balance (LA-TB) objective effectively neutralizes structural length biases. The authors successfully proved that LA-TB is a minimum-distortion I-projection, preserving ~91% of original mode rankings.
* Strong Empirical Performance: PowerFlow consistently outperforms existing unsupervised (RLIF) baselines and shows superior sample efficiency compared to supervised methods (GRPO) on 1.5B/3B scale models.
* Dual-Nature Control: The framework offers a unified control knob (alpha) that achieves Pareto improvements in creative tasks—enhancing both diversity and quality simultaneously.

While initial concerns regarding baseline fairness and ranking distortion were raised, the authors' rebuttal provided comprehensive proofs and significance tests that satisfied all reviewers. The work is technically sound and highly relevant to the ICML community.